# Chromatin modifier MTA1 regulates mitotic transition and tumorigenesis by orchestrating mitotic mRNA processing

Jian Liu [1,2,8], Chunxiao Li[1,3,8], Jinsong Wang[1,8], Dongkui Xu[4,8], Haijuan Wang[1], Ting Wang[1], Lina Li[2], Hui Li[1], Peng Nan[1], Jingyao Zhang[1], Yang Wang[5], Changzhi Huang[1], Dong Chen [6], Yi Zhang [6], Tao Wen [2✉], Qimin Zhan [1,7✉], Fei Ma [3✉] & Haili Qian [1✉]

Dysregulated alternative splicing (AS) driving carcinogenetic mitosis remains poorly understood. Here, we demonstrate that cancer metastasis-associated antigen 1 (MTA1), a well-known oncogenic chromatin modifier, broadly interacts and co-expresses with RBPs across cancers, contributing to cancerous mitosis-related AS. Using developed fCLIP-seq technology, we show that MTA1 binds abundant transcripts, preferentially at splicing-responsible motifs, influencing the abundance and AS pattern of target transcripts. MTA1 regulates the mRNA level and guides the AS of a series of mitosis regulators. MTA1 deletion abrogated the dynamic AS switches of variants for *ATRX* and *MYBL2* at mitotic stage, which are relevant to mitosis-related tumorigenesis. MTA1 dysfunction causes defective mitotic arrest, leads to aberrant chromosome segregation, and results in chromosomal instability (CIN), eventually contributing to tumorigenesis. Currently, little is known about the RNA splicing during mitosis; here, we uncover that MTA1 binds transcripts and orchestrates dynamic splicing of mitosis regulators in tumorigenesis.

[1] State Key Laboratory of Molecular Oncology, National Cancer Center/National Clinical Research Center for Cancer/Cancer Hospital, Chinese Academy of Medical Sciences and Peking Union Medical College, Beijing 100021, China. [2] Medical Research Center, Beijing Chao-Yang Hospital, Capital Medical University, Beijing 100020, China. [3] Department of Medical Oncology, National Cancer Center/National Clinical Research Center for Cancer/Cancer Hospital, Chinese Academy of Medical Sciences and Peking Union Medical College, Beijing 100021, China. [4] VIP Department, National Cancer Center/National Clinical Research Center for Cancer/Cancer Hospital, Chinese Academy of Medical Sciences and Peking Union Medical College, Beijing 100021, China. [5] Institute of Cancer Stem Cell, Dalian Medical University, Dalian 116044, China. [6] Center for Genome Analysis, ABLife Inc, Wuhan 430075, China. [7] Key laboratory of Carcinogenesis and Translational Research (Ministry of Education/Beijing), Laboratory of Molecular Oncology, Peking University Cancer Hospital & Institute, Beijing 100142, China. [8] These authors contributed equally: Jian Liu, Chunxiao Li, Jinsong Wang, Dongkui Xu. ✉email: wentao5281@163.com; zhanqimin@bjmu.edu.cn; drmafei@126.com; qianhaili001@163.com

Cell cycle dysregulation is essential for cellular transformation. Dysregulation of the mitotic transition gives rise to aberrant mitosis, CIN and resultant tumorigenesis and progression[1,2]. Mitotic progression is conventionally thought to be achieved by regulation of mitosis regulators at multiple levels, like transcription, translation, posttranslational modification, protein stability and protein-protein interactions, whereas RNA processing is considered to be globally suspended during mitosis[3,4]. However, emerging evidences suggest that the whole cell cycle, including the mitotic stage, is functionally coupled with pre-mRNA processing[4], and mitotic RNA processing contributes to spindle assembly in Xenopus[5]. Unfortunately, currently, little is known about how pre-mRNA processing, notably AS, is regulated during mitosis.

MTA1 was found metastasis-related in adenocarcinoma cell lines from breast cancer[6], and represents an important component of the nucleosome remodeling and deacetylation complex (NuRD). MTA1 accesses nucleosomes and represses DNA-dependent transcription by coupling with HDAC through its ELM2-SANT domains, in favor of oncogenesis and cancer progression[7].

We previously validated presence of cytoplasmic MTA1 and its positive correlation with tumor progression[8,9]. We also described a periodic MTA1 distribution pattern during the cell cycle[10]. MTA1 is disassociated with the chromatin from the beginning of prophase and free of chromosome-binding in the cytoplasm across metaphase and anaphase[10]. These suggest molecular mechanisms beyond chromatin modification.

Here, we demonstrate MTA1, as an RNA-binding chromatin-associated protein (CAP), facilitates cancer development and progression by governing transcript abundance and AS pattern of mitosis regulators during mitosis.

## Results

**MTA1 is coexpressed and interacts with RBPs in cancers.** Functional genes and their products are coordinately expressed in temporal and spatial patterns to orchestrate certain cellular activity[11], which is often indicative of their biological functions[12]. On this concept, we built an *MTA1*-centralized transcriptional coexpression network using RNA-seq data of 382 colorectal adenocarcinoma (COAD) samples from The Cancer Genome Atlas (TCGA). Functional annotation of the 657 *MTA1*- correlated genes (criteria: abs (PCC) > = 0.3) highlighted the known DNA-related functions of MTA1, such as DNA repair, DNA-dependent transcription, etc.[13], and the unknown functions in RNA-related processes, especially RNA splicing and mRNA processing (Supplementary Fig. 1a), which was extendedly confirmed in the 32 cancer types in TCGA (Fig. 1a), strongly indicative of MTA1's role in RNA processes.

Further validation in the pan-cancer CCLE cell line samples (n = 967) and pan-cancer patient derived xenografts (PDX) samples (GSE78806, n = 661) showed high enrichment on RNA processes at the very top (Supplementary Fig. 1b, c). RNA-related enrichment of MTA1's function was largely attributed to its consistent correlation with numerous RBPs across different cancers (Supplementary Fig. 1d, some examples are shown). All these support the idea that MTA1 cooperates with RBPs to orchestrate RNA processes.

The coexpressed genes act through signaling cascades or physical interactions to orchestrate a certain biological process[11,12,14]. MTA1 deletion didn't cause remarkable changes in the coexpressed RBP mRNA levels in the RNA-seq assay, rising the speculation of physical interaction between MTA1 and RBPs to coordinate RNA-related processes. The fact that most components of NuRD correlated significantly with *MTA1* in the CCLE pan-cancer samples (Supplementary Fig. 1e), verified that interacted molecules emerged in parallel at the transcriptional level.

To examine the possible MTA1 interaction with RBPs, we performed coimmunoprecipitation (co-IP)-coupled mass-spectrometry arrays in HCT116 cells using two MTA1 antibodies for reciprocal validation (Fig. 1b). Ninety-five proteins were cocaptured by both antibodies (232 proteins for mouse anti-MTA1 antibody and 136 for rabbit, respectively, Supplementary Data 1, Supplementary Fig. 1f). Besides the core NuRD components like MTA1, HDAC1/2, RBBP7/4, MTA2, GATAD2A/B, MBD3 and CHD4 (Fig. 1c, Supplementary Data 1), RBPs constituted the majority of MTA1 potential interactors (54.3% for the mouse antibody, 126/232; 50.7% for the rabbit antibody, 69/136; and 60.0% for overlapped proteins, 57/95). These 57 RBPs were enriched on multiple RNA processes (Fig. 1d), highly consistent with those from co-expression analysis. Of the 57 MTA-binding RBPs, 38 were available for transcriptional level from the CCLE microarray data, with 92.11% (35/38) significantly correlated with MTA1 ($p < 0.05$), including *PTBP1* ($r = 0.4715$), *SFPQ* ($r = 0.443$), and *HNRNPM* ($r = 0.3842$). These data suggest that MTA1 contributes to RNA processes by temporally and spatially cooperating with RBPs.

These MTA1-interacting RBPs included a number of spliceosome components and pre-mRNA splicing regulators (Fig. 1c, Supplementary Data 1). Many RNA recognition motif (RRM)-containing proteins were also coimmunoprecipitated, including NONO, HNRNPM, SFPQ, HNRNPA2B1, G3BP1, PTBP1, HNRNPD, NCL, HNRNPA1 and HNRNPAB (Supplementary Data 1). The direct interaction between RBPs and MTA1 was validated by co-IP analyses, using HDAC2 as a positive control (Fig. 1e, Supplementary Fig. 1g). Components of the SMN-Sm protein complex (SMN, GEMIN3, GEMIN4, GEMIN7 and STRAP) involving the spliceosomal snRNP assembly[15] were confirmed to interact with MTA1 by co-IP (Fig. 1e).

We have reported the presence of cytoplasmic MTA1[8]. Here, we observed abundant granular MTA1 resemble to the RBP-formed RNP granules in the cytoplasm of cells HCT116 (Supplementary Fig. 1h), NCI-H446, SF-767, HEK293 and Caski[9]. Co-localization assays demonstrated a proportion of cytoplasmic MTA1 granules were colocalized to the SMN granules (Supplementary Fig. 1i, Supplementary Fig. 1j) and the colocalization could be enhanced by UV irradiation (Fig. 1f), indicating a possible role of MTA1 to affect alternative splicing by modulating spliceosomal SNP maturation. Moreover, some cytoplasmic MTA1 were also colocalized to RPS3 (40 S ribosomal protein S3) (Supplementary Fig. 1k) and YBX1 RNP granules (Fig. 1g). However, MTA1 and PTPB1 colocalization was observed exclusively in the nucleus (Supplementary Fig. 1l). These cytoplasmic RNP granules were distributed along the microtubule and microfilament cytoskeletons from the center region to the leading edge of the cell (Supplementary Fig. 1m), consistent to the cytoskeleton-guided transportation of RNP granules[16].

To exclude that MTA1 interaction with RBPs may be mediated by an RNA bridge[17], we conducted coimmunoprecipitation assays with or without DNase I and RNase A. The treatment did not reduce the levels of RBPs immunoprecipitated by MTA1, instead slightly enhanced MTA1 binding to SFPQ, NONO and PTBP1 (Supplementary Fig. 1n). These results suggest that the MTA1-RBP interaction may not rely on RNA bridge.

**MTA1 binds to splicing-responsible motifs in pre-mRNA.** Extensive MTA1 interaction with RBPs strongly suggested that MTA1 interacts with RNA transcripts. To verify this hypothesis,

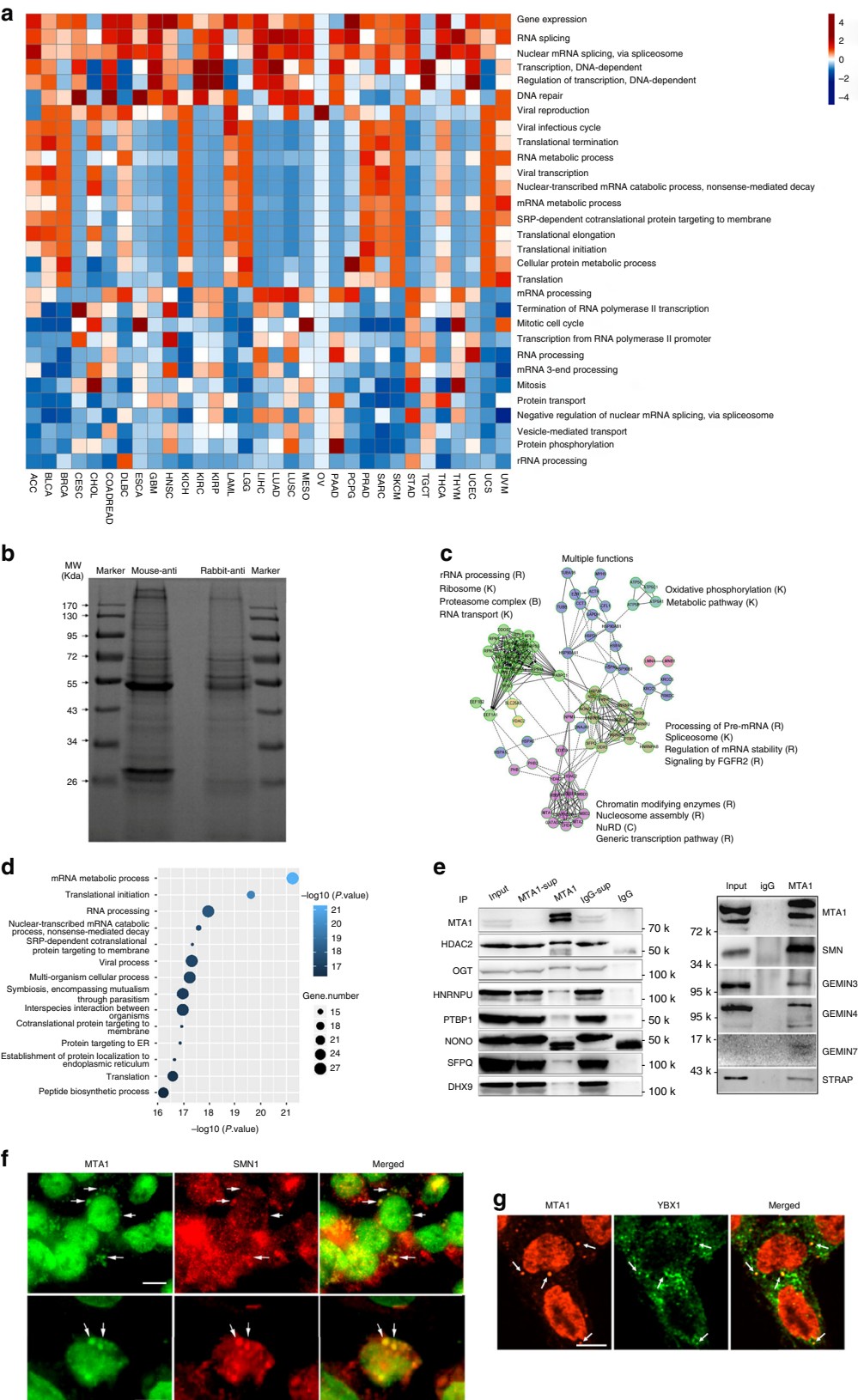

we developed a formaldehyde crosslinking immunoprecipition (fCLIP)-coupled high-throughput sequencing method by combining the advantages of formaldehyde RNA immunoprecipitation (fRIP)[18] and crosslinking immunoprecipitation (CLIP) techniques to profile the RNA targets bound by the MTA1

complex in HCT116 cells (Fig. 2a). MTA1-fCLIP reads were extensively distributed in the genic regions after mapping to the human genome (GRCH38) by TopHat2[19], indicating a global association of MTA1 with transcripts (Supplementary Fig. 2a). MTA1-fCLIP reads occupied dominantly the CDS region,

**Fig. 1 MTA1 coexpresses and interacts with RBPs in cancer. a** Comprehensive functional landscape of the MTA1 coexpressing genes generated from a panel of 32 TCGA transcriptome datasets revealing a consistent enrichment in RNA processes. Functional enrichment analysis was performed using DAVID 6.8 database. **b** SDS-PAGE separation of the MTA1 interacting proteins captured by mouse- and rabbit- specific antibodies against MTA1 in HCT116 cells. **c** Network presentation of the functional clusters enriched by the MTA1 interacting proteins. A reactome analysis was performed for the MTA1 interacting proteins identified by both antibodies. **d** A functional enrichment analysis of the overlapping 57 RBPs using DAVID 6.8 database. **e** Co-IP and Western blot identification analyses of the MTA1 interacting RBPs. The HDAC2 and OGT coimmunoprecipitation with MTA1 is also shown. Sup, supernatant. **f** Immunofluorescence co-localization of MTA1 (green) and SMN (red) after UV irradiation. **g** Immunofluorescence co-localization of MTA1 and YBX1. Results in **b**, **e**–**g** are representative of three independent repeats with similar results. Scale bar = 10 μm for **f** and **g**.

followed by the intronic and 3′UTR regions (Fig. 2b). However, compared to the RNA-seq reads from polyadenylated mature mRNAs, the MTA1-fCLIP reads were less abundant in the CDS region and more enriched on the intronic region (Fig. 2b), demonstrating that MTA1 bound to nascent transcripts prior to intron splicing. More than one-third (5169) of intronic peaks (14,084, 25.16% of total 55,975) were located at the splice junctions bridging the exon and intron.

To profile MTA1 binding feature, we mapped the MTA1-fCLIP reads and RNA-seq reads to the 5′ and 3′ splice sites of the internal exons. The MTA1-fCLIP reads were densely enriched in the internal exons, similar to the RNA-seq reads, regardless of the presence or absence of the fCLIP peak (Fig. 2c). The MTA1-fCLIP reads from peak-containing mRNAs more densely spanned the splice sites than the RNA-seq reads and the fCLIP nonpeak reads (Fig. 2c, boxed), suggesting that MTA1 occupied the 5′ and 3′ splice sites.

We next enriched the MTA1-binding motifs by Homer[20] and found that the 5′ splicing site consensus AGGUAAG was always among the most enriched motifs in all fCLIP peaks called by ABLIRC[21], Piranha[22] and CIMS[23] (accounting for 33.77%, 34.25% and 20.00% of all binding peaks, respectively) (Supplementary Fig. 2b and Fig. 2d, e). A U-rich motif, upstream of the 3′ splice site, was also enriched at the exon-intron boundaries (Supplementary Fig. 2b, c). Besides, MTA1 also preferentially bound to the GAAGAA motif (Supplementary Fig. 2b and Fig. 2f), a well-known exonic splicing enhancer (ESE) for pre-mRNA splicing[24]. Another MTA1-binding motif was UGGAC (Supplementary Fig. 2b and Fig. 2g), a highly conserved N6-methyladenosine (m6A) modification motif important for mRNA stability and splicing[25,26]. Above evidences suggest that MTA1 may regulate AS by binding to splicing elements.

To confirm the reliance of MTA1 on these specific motifs to bind mRNA, we categorized the MTA1-bound mRNAs by the number of the above four motifs and found that the binding abundance increased with motif occurrences, further indicating that MTA1 specifically bound to these motifs (Fig. 2h).

The MTA1-fCLIP genes were enriched on cancer pathways, cell cycle, ubiquitin-mediated proteolysis, endocytosis, etc. (Supplementary Fig. 2d). By RIP-qPCR verification, 17 of the 20 candidates showed enrichment at the MTA1-binding sites, including cancer invasiveness meditators *ITGB4* and *CD151*[27,28] (Fig. 2i and Supplementary Fig. 2e).

**MTA1 binding affects the level and splicing of transcripts.** RBPs could post-transcriptionally affect gene expression[29]. Therefore, we first determined whether MTA1-binding influenced the abundance of target mRNAs. By CRISPR-Cas9-mediated MTA1 knockout (CrMTA1) (Supplementary Fig. 3a) and RNA-seq, MTA1-fCLIP targets with higher-affinity (219 upregulated vs. 74 downregulated, FDR < 0.01) showed more significantly upregulated by MTA1 deletion (Fig. 3a). These suggest that MTA1 globally destabilizes its target mRNAs.

MTA1 deletion generated 1510 differentially expressed genes (DEGs) (FDR < 0.01) (Fig. 3b and Supplementary Data 2), which enriched on previously disclosed p53, HIF-1, NF-kappa B and PI3K-Akt pathways (Supplementary Fig. 3b). The majority of the DEGs (1187/1510, 78.61%, Fig. 3c, *p*-value = 3.73e − 246, hypergeometric test) transcripts physically interacted with MTA1 at regions covering the CDS, 3′UTR, boundary, intron and 5′UTR.

MTA1 is known to regulate transcription by promoter binding[30,31]. Thus, we compared the mRNA-binding and promoter-binding contributions of MTA1 to gene expression. ChIP-seq assay showed that MTA1 reads were dense around the transcription start site (TSS) (Supplementary Fig. 3c), confirming its association with promoters. A parallel MTA1 ChIP-seq data (GSE91687) from MCF-7 breast cancer cell line in the ENCODE project[32] generated significant overlap with our data from HCT116 in promoter-binding genes (−5K to 2 K relative to TSS) (Supplementary Fig. 3d, *p*-value = 5.62e − 88, hypergeometric test). However, we did not detect a significant difference in the transcript abundance among high-, median-, and low-affinity ChIP targets (Fig. 3d). Consistently, only 25.10% (379/1510) of the DEGs showed promoter-binding by MTA1 (Fig. 3e). These results suggested that MTA1's promoter binding acts more subtly than its mRNA-binding mechanism on transcript levels.

The physical interaction with splicing factors and binding preference to mRNA splicing-related motifs, suggested a function of MTA1 in mRNA splicing. Using the alternative splicing detection tool ABLas[21], we found that MTA1 deletion globally influenced the AS of its bound transcripts, which correlated well with the RNA-binding affinity (Fig. 3f), while no significant AS alterations were found correlated with promoter-binding affinity (Fig. 3g).

MTA1 knockout led to 1467 significantly regulated AS events (RASEs, *p*-value < 0.05) residing in 1230 genes (RASGs) (Fig. 3h and Supplementary Data 3), and most of the RASGs (1026/1230, 83.41%) showed MTA1-binding with their transcripts (Fig. 3i, *p*-value < 2.2e − 16). MTA1 deletion generated more IR events, while other ASs were equivalent in number (Fig. 3h, *p*-value = 4.494e − 05, Fisher's exact test), indicating that MTA1 facilitated intron splicing. We selected 6 IR and 12 non-IR events for qRT-PCR validation, 16 of which (88.89%, 16/18) were consistent with the sequencing results with 12 (75%, 12/16) were significant (*p*-value < 0.05, *t*-test) (Fig. 3j and Supplementary Fig. 3e). The *CTTN* mRNA, as an identified MTA1-bound example (Supplementary Fig. 2e), was verified for AS pattern switch after MTA1 knockdown by Crispr-cas9 or shRNA (Supplementary Fig. 3f).

Out of the 1026 MTA1-bound RASGs, 113 (11.01%) were concomitantly changed in mRNA abundance (*p*-value = 6.49e-19, hypergeometric test) (Supplementary Fig. 3g). LTK, another MTA1-fCLIP targets, is a ros/insulin receptor tyrosine kinase that controls cell growth and differentiation[33,34]. MTA1 deficiency upregulated *LTK* mRNA levels (Supplementary Fig. 3h) and promoted the inclusion of its 183-nt exon 7 (Supplementary Fig. 3i, j), which encodes a predicted glycosylation site. These

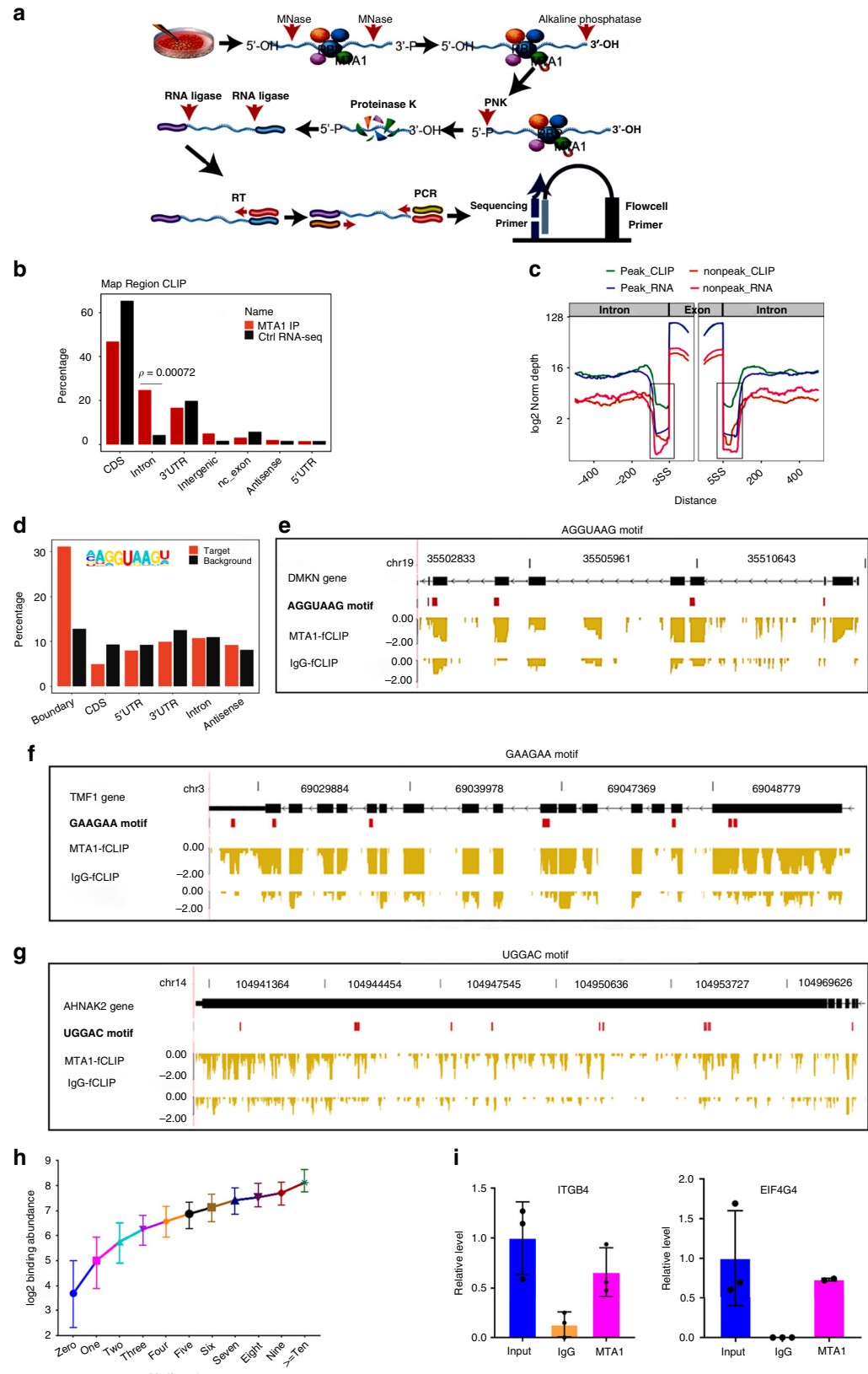

results indicate that MTA1-mRNA association influences both the abundance and AS pattern of target transcripts.

**MTA1-transcript association regulates mitotic transition**. To explore the RNA binding-related role of MTA1 in cancer, we enriched the MTA1-fCLIP DEGs and RASGs using DAVID. Both the 1188 MTA1-fCLIP DEGs and the 1026 MTA1-fCLIP RASGs were enriched on mitosis-related biological processes (Supplementary Fig. 4a, b). In comparison, the DEGs with MTA1-binding on the promoter showed enrichment mainly on

**Fig. 2 Identification of the MTA1-associated RNAs using fCLIP-seq. a** Flow diagram showing the key steps in the fCLIP-seq experiment to identify the MTA1 complex-associated RNAs in the HCT116 cells. **b** Bar plotting the genomic region distribution of the uniquely mapped reads from the MTA1 fCLIP-seq and RNA-seq samples. Significant enrichment was observed in the intronic region. $p$-value = 0.00072, Fisher's exact test, two-tailed. **c** fCLIP-seq and RNA-seq read density around the boundaries between internal exons (100 nt) and adjacent introns (400 nt). The genes were divided into the MTA1 fCLIP peak containing or lacking groups. The normalization was carried out by dividing the accumulated reads (uniquely aligned) in each group of genes by the gene numbers. The RNA-seq and CLIP-seq signals were then normalized using the maximum value from the four groups. **d** The frequencies of the 5′ splice site AGGUAAG motif in the peaks separated by regions. **e** Example illustration showing the binding preference of MTA1 to the AGGUAAG motif in *DMKN*. **f** Example illustration showing the binding preference of MTA1 to the GAAGAA motif in *TMF1*. **g** Example illustration showing the binding preference of MTA1 to the TGGAC motif in *AHNAK2*. **h** The MTA1-bound mRNAs were categorized into 11 groups by the numbers of motifs on the manuscript, and the mRNA binding activity increased with the occurrence of the specific motifs. Data are represented as median with the interquartile range. **i** RIP-PCR validation of the MTA1-associated *ITGB4* and *EIF4G4* transcripts ($n = 3$ per group except EIF4G4 MTA1 $n = 2$). Data are represented as mean ± SD.

previously disclosed processes, such as apoptotic process, endothelial cell differentiation, cell proliferation, and cell motility (Supplementary Fig. 4c). These indicate that MTA1 may use DNA- and RNA-related mechanisms to execute distinct functions, and MTA1 may regulate mitosis via mRNA-associated posttranscriptional regulation. Besides, the top MTA1-correlated genes generated from the public single-cell RNA-seq of HCT116 cells (GSE51254) also strongly pointed to RNA splicing and mitosis transition (Supplementary Data 4).

We assessed MTA1 influence on mitotic transition in response to spindle damage by nocodazole, which induces mitotic arrest, using flow cytometry in HCT116 sublines with MTA1 stable-overexpression or knockdown (Fig. 4a). Forced MTA1 expression in HCT116 cells (HCT116-OE-3) significantly decreased the level of mitotic arrest induced by spindle damage, whereas MTA1 knockdown (HCT116-KD) increased the level of arrest, as measured by the DNA content (Figs. 4b, c) or mitosis specific marker p-H3 staining (Figs. 4d, e). MTA1-overexpressing cells also showed lower mitotic arrest-induced cell death after 24 h nocodazole treatment (Supplementary Fig. 4d). Morphologically, MTA1-overexpressing cells largely maintained normal shape and were adherent after nocodazole treatment, while MTA1-knockdown cells became round and floating (Supplementary Fig. 4e). This tendency was confirmed in EGFP-MTA1-overexpressed HCT116 cells (Fig. 4f) and MTA1-stably over-expressed KYSE-410 cells (Fig. 4g). These data reveal that MTA1 forces mitotic transition in response to mitotic damage in cancer cells.

**MTA1 posttranscriptionally regulates mitotic transcriptome.** To determine how MTA1 modulates the mitotic transition, we examined the effect of MTA1 on the sequential transcriptomes that were synchronized at G1/S by double-thymidine block and released into mitosis. According to the expression patterns of the mitosis-entering marker CDK1 (Supplementary Fig. 5a) and other specific mitosis markers, such as CCNB1, CCNB2, CENPF, KIF2C and AURKA (Supplementary Fig. 5b), we grouped the sequential samples into three subsets spanning pre-mitosis, mitosis and post-mitosis (Supplementary Fig. 5b).

MTA1 imposed a global destabilization function on the higher-affinity fCLIP target transcripts during mitosis, and preferentially on bound mitosis-associated gene transcripts (genes under the GO term mitotic cell cycle, GO: 0000278) (Supplementary Fig. 5c, left). MTA1 deletion also caused a more obvious AS pattern alteration in fCLIP targets with higher affinity (Supplementary Fig. 5c, right), but no significant correlation between MTA1 promoter-binding affinity and mRNA abundance or AS were found in the mitotic cells (Supplementary Fig. 5d). Consistently, 543 (77.96%) of MTA1 deletion-induced 735 DEGs ($p < 0.01$) in mitotic cells were MTA1-bound in the fCLIP-seq data. Of the 165 (22.45% of total) DEGs that were MTA1-promoter bound in the ChIP-seq data, most (128) were also bound by MTA1 in their

transcripts. Thus, the mere promoter binding-sensitive DEGs made up only 5.03% (37/735). When filtrated by $p < 0.05$, 1422 DEGs were generated, and the results were very similar, with 77.29% (1099/1422) bound by MTA1 in their transcripts, 22.78% (324/1422) bound by MTA1 on their promoters, and 4.15% (59/1422) mere promoter binding (Supplementary Fig. 5e). For RASEs, we identified 2491 RASEs in 1798 RASGs (FDR < 0.01) in the mitotic cells, 85.26% (1533/1798) of which were MTA1-bound by fCLIP and only 2.50% (45/1798) showed mere promoter association with MTA1 (Supplementary Fig. 5f). These data indicate MTA1 modulates the mitotic transcriptome largely in an mRNA association-dependent manner.

Of the 1422 DEGs ($p < 0.05$) identified in the mitotic cell samples by MTA1 deletion, 1133 were not significant by $p \leq 0.05$ in the premitotic cells. Similarly, 903 of the 1798 RASGs (FDR < 0.01) identified from the mitotic cells were not significant by $p \leq 0.05$ in the premitotic cells, indicating that most of these posttranscriptional events are mitosis-specific (some examples are discussed below). As MTA1 is extrachromosomal and deprived of transcriptional regulation at most mitotic duration[10], it is also reasonable to attribute the MTA1-driven mitosis-specific transcriptome alteration to posttranscriptional regulation.

**MTA1 alters transcript abundance of mitosis regulators.** DAVID clustering of the DEGs from synchronized mitotic cells highlighted mitosis-related processes (Supplementary Fig. 6, Supplementary Data 5), and supported the role of MTA1 in mitotic transition. Of the 1422 mitotic DEGs, at least 208 were essential mitosis regulators, 188 (90.38%) of which with MTA1-binding to their transcripts. To describe the effect of MTA1-mRNA binding on mitosis regulators, we examined TTK, ZWILCH, DLG1, TERF1, CENPF, CNOT7, SMARCA4, CENPJ, SGO1, ZWINT, LIG1, KIF23, RANBP1, USP47 and UBA52, whose transcripts are bound by MTA1 (Fig. 5a). Besides the consistent conspicuous abundance alteration (Fig. 5b, Supplementary Fig. 7a), many of these regulators, such as TTK, ZWILCH, DLG1, TERF1 and CENPF, were specifically regulated at the mitotic stage but not at pre-mitosis (Fig. 5b), indicating MTA1 exerted a mitosis-specific modulation on these mitosis regulators. Most of these regulators were upregulated at mitosis after MTA1 deletion, validating again the role of MTA1 in destabilizing its target transcripts (Fig. 5b, Supplementary Fig. 7a). Cell cycle regulators are expressed periodically on demand of the cell cycle stages[35,36]. MTA1 also changed and even reversed the periodic expression of mitotic regulators *CNOT7* (Fig. 5b) and *LIG1* (Supplementary Fig. 7a).

As for how MTA1 forced the mitotic transition upon spindle damage, we found MTA1 deletion upregulated the main components and activators of the spindle assembly checkpoint (SAC), whose transcripts were bound by MTA1, including MAD1 and MAD2, CENPF, TTK, ZWLCH, CENPJ, ZWINT, RANBP1, SGO1, etc. We validated the mitosis-specific upregulation of *TTK*,

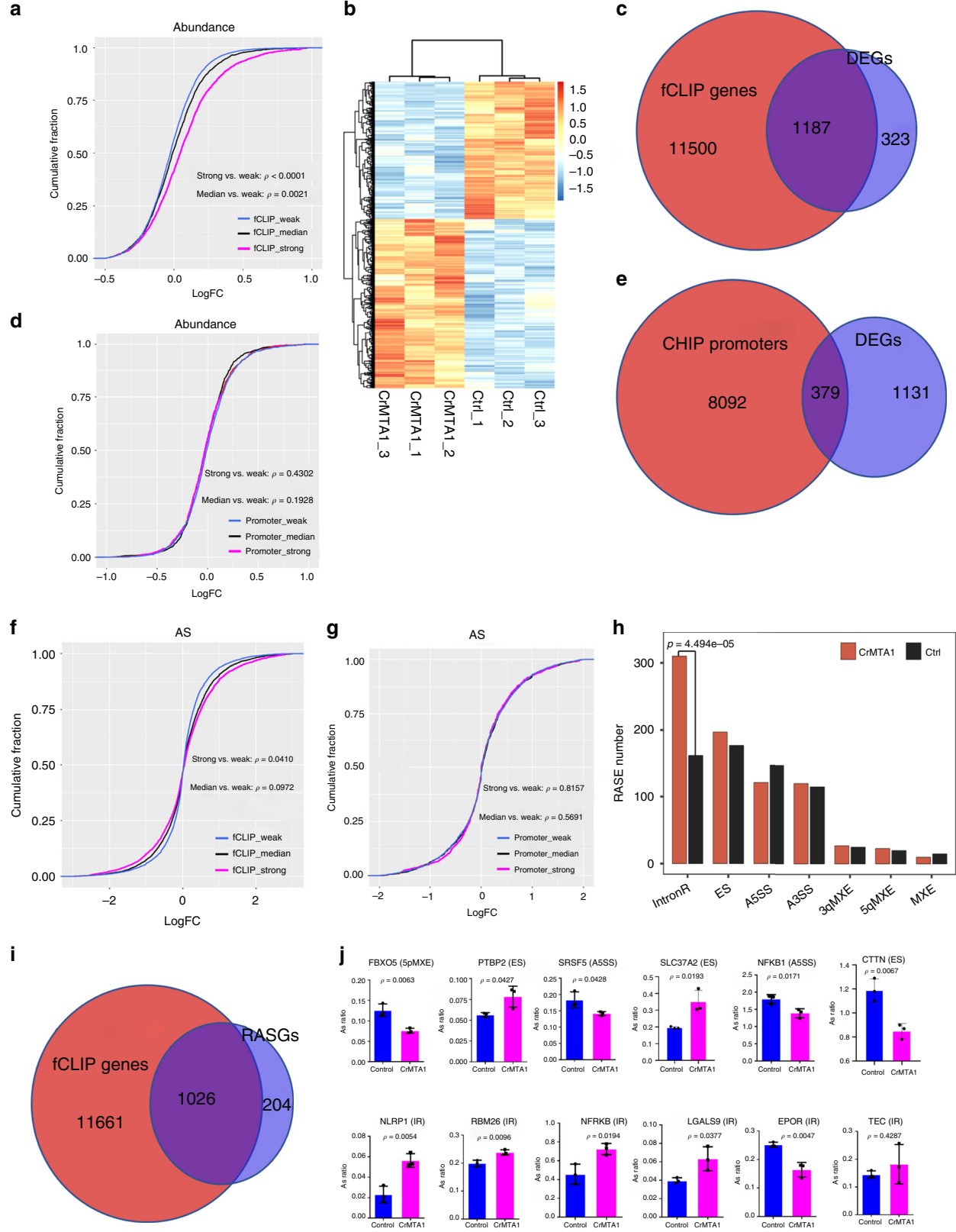

 *ZWILCH*, *SOG1* and *DLG1* by MTA1 deletion using qPCR (Supplementary Fig. 7b). Deletion of TTK, SOG1 or DLG1 but not ZWILCH contributes to a partial recovery of the mitosis transition ability of MTA1-knockout HCT116 cells (Fig. 5c, Supplementary Fig. 7c). Hierarchical assembly of SAC components, such as MAD1, MAD2 and CENPF, ensures the activation

of SAC. MTA1 deletion upregulated the level of MAD1 and MAD2 transcripts and proteins as well as their mutual interaction (Supplementary Fig. 7d, e).

**MTA1 drives mitosis-specific AS switch of mitosis regulators.** DAVID clustering of the mitotic RASGs also enriched on mitosis-

**Fig. 3 MTA1 influences mRNA abundance and alternative splicing. a** The fCLIP target transcripts were classified as strong-, median- or weak-affinity with MTA1 according to the fCLIP-seq data normalized by the RNA-seq expressional abundance in the control. The empirical cumulative distribution function of the log2 CrMTA1/control fold changes of the abundance for each set were plotted. **b** Hierarchical clustering heatmap of the differentially expressed genes (DEGs) between CrMTA1 and the control. **c** Venn diagram showing the overlap between the MTA1 binding transcripts (fCLIP genes) and DEGs. **d** The ChIP promoters were classified as strong-, median- or weak-affinity with MTA1. The affinity was defined using a binding affinity score, which is the sum of fold enrichment (MTA1-ChIP/Input) of peaks in the gene promoter region according to the ChIP-seq data. The empirical cumulative distribution function of the log2 CrMTA1/control fold changes of the abundance for each set were plotted. **e** Venn diagram showing the overlap between the MTA1 binding gene promoters (ChIP-promoters) and DEGs. **f** The fCLIP target transcripts were classified as strong-, median- or weak-affinity with MTA1 according to the fCLIP-seq data normalized by the RNA-seq expressional abundance in the control, and the empirical cumulative distribution function of the log2 CrMTA1/control fold changes of the AS ratio for each set were plotted. **g** The ChIP promoters were classified as strong-, median- or weak-affinity with MTA1, and the empirical cumulative distribution function of the log2 CrMTA1/control fold changes of the AS ratio for each set were plotted. **h** Bar plot showing that MTA1 deletion generated more IR events. IntronR, intron retentions; ES, exon skips; MXE, mutually exclusive exons; A5SS, alternative 5′ss; A3SS, alternative 3′ss. **i** Venn diagram showing that the transcripts of most of the RASGs were bound by MTA1. **j** RT-qPCR validation for the non-IR and IR events. Data are represented as mean ± SD ($n = 3$ per group). **a**, **d**, **f** and **g** One-way ANOVA with Tukey's multiple comparisons test. **h** Two-tailed Fisher's exact test. **j** Two-tailed Student's $t$-test.

related functions (Supplementary Fig. 8a, Supplementary Data 6). Of the above identified RASGs, at least 248 were mitosis regulators, and 225 (90.73%) of which showed MTA1-binding to their transcripts. ATRX is a SWI/SNF-like chromatin remodeler localized to pericentromeric heterochromatin[37]. It is hyperphosphorylated at the onset of mitosis[38], and plays an essential role in regulating mitosis[39]. There are multiple splicing variants for ATRX, of which the ATRX variant 1 and variant 2 are two major forms by retention or skipping of exon 6[40] (Fig. 6a). MTA1 binds to ATRX transcripts at sequences flanking exon 6 and exonic ESE motifs according to the fCLIP-seq data (Fig. 6b). The two ATRX variants were mutually exclusive across the cell cycle, both in the control and MTA1-depleted cells (Fig. 6c, lower two). MTA1-deletion reversed the ratio between ATRX variant 1 and variant 2, exhibiting a shift in the AS pattern from exon retention to skipping (Fig. 6c, upper two). The MTA1 deletion-caused level divergence of both variants started specifically at the onset of mitosis, indicating that the regulation was mitosis-specific (Fig. 6c, upper two).

Just as the mRNA level fluctuation of the cell cycle regulators during the cell cycle[35,36], here, we disclosed that the AS pattern of many mitosis regulators was also dynamically shifted during the cell cycle, e.g., ATRX (Fig. 6c, lower, and Fig. 6d), and the following discussed MYBL2, and MTA1 dynamically shifted the AS spectrum of these mitosis regulators during mitosis. MTA1 deletion did not notably alter the abundance of ATRX pre-mRNA, but diminished the fluctuation of pre-mRNA AS kinetics after entering into the mitotic stage without affecting the premitotic AS pattern (Fig. 6c, shadow, and Fig. 6d). This evidenced that MTA1 exquisitely coordinated the constant conversion of the AS pattern of mitosis regulators to drive mitosis progression.

The MYB Proto-Oncogene Like 2 (MYBL2, or B-MYB), is a key regulator in orchestrating the G2/M cell cycle transition, mitotic spindle functioning and mitosis fidelity[41–43]. It produces two transcript variants by alternative exon 3 skipping (Fig. 6e). MTA1 bound to its sequences flanking exon 3 and multiple TGGAC m6A motifs (Fig. 6f). MTA1 deletion switched the MYBL2 pre-mRNA splicing from exon 3 skipping to retention, which was specific before mitotic entry and during mitosis, and disappeared upon mitotic exit (Fig. 6g). Notably, MTA1 deletion abrogated the MYBL2 AS fluctuation at mitosis (Fig. 6g, h). The dysregulation of above AS events in ATRX and MYBL2 are significantly related to mitosis (Supplementary Fig. 8b) and tumorigenesis (Supplementary Fig. 8c) in cancer patients, suggesting that MTA1 may regulate mitosis and tumorigenesis by targeting the AS of these two mitosis regulators.

The mitosis-specific AS alteration of MYBL2 pre-mRNA was validated using qPCR (Supplementary Fig. 8d). Moreover, forced expression of MYBL2 variant 2 contributed more than variant 1 to overcoming of the nocodazole-induced mitotic arrest in MTA1-knockout HCT116 cells, as evidenced by high content analysis system (Supplementary Fig. 8e, f) and flow cytometry (Supplementary Fig. 8g) assays, suggesting that MTA1 may drive mitotic transition partially by mitotically modulating MYBL2 AS pattern. MTA1 also imposed an AS regulation on CNOT1, specifically at late mitosis (Supplementary Fig. 8h, i). For CDC25C, MTA1 deletion changed the dominant variant CDC25C-203 (Ensembl name) in the control cells to CDC25C-207 (on mitosis entry) and CDC25C-209 (mitosis), depending on the cell cycle stage (Supplementary Fig. 8j).

Collectively, MTA1 controlled the AS pattern and kinetics of mitosis regulators in a cell cycle stage-dependent manner, which, along with the mRNA abundance regulation, comprehensively contributed to mitotic transition.

**MTA1 leads to CIN and tumorigenesis.** Defective mitotic arrest leads to misaligned chromosomes and aberrant chromosome segregation[2]. MTA1 overexpression-induced defective mitotic arrest caused obvious mitotic abnormalities, including chromosome lagging, bridging, misalignment and multipolarized chromosomes in cancer cells (Fig. 7a, b, Supplementary Fig. 9a).

CIN is common in solid tumors and originates from aberrant chromosome segregation during mitosis[2]. Thus, we determined whether MTA1-induced mitotic defectiveness resulted in CIN. MTA1 overexpression caused an increased proportion of multinucleated and micronucleated cells (manifestations for aneuploidy) (Fig. 7c, d and Supplementary Fig. 9b), and Crispr-cas9-mediated MTA1-knockout led to a reduction in both spontaneous and nocodazole-induced micronuclei ratio in HCT116 (Supplementary Fig. 9c, d). Likewise, flow cytometry assays showed less aneuploidy (>4n) in MTA1-knockout KYSE410 cells on nocodazole treatment (Supplementary Fig. 9e, f).

Mitosis abnormality usually reflects malignancy of clinical cancers. We next detected the correlation between MTA1 and mitotic defectiveness in clinical cancer specimens. MTA1 was significantly higher in colon cancers than in their adjacent counterparts (Supplementary Fig. 9g). In the 180 cases of paired tissues, 0 (0%) were found with pathological mitosis in adjacent normal, while 141 (78.33%) in cancer, and the frequency of pathological mitosis in cancer increased with MTA1 intensity scaling-up ($p < 0.001$, Fig. 7e, f, Supplementary Table 1).

CIN drives oncogenesis and progression[1,2]. MTA1 overexpression caused an obvious pleura invasion (3/6, 50%) and more

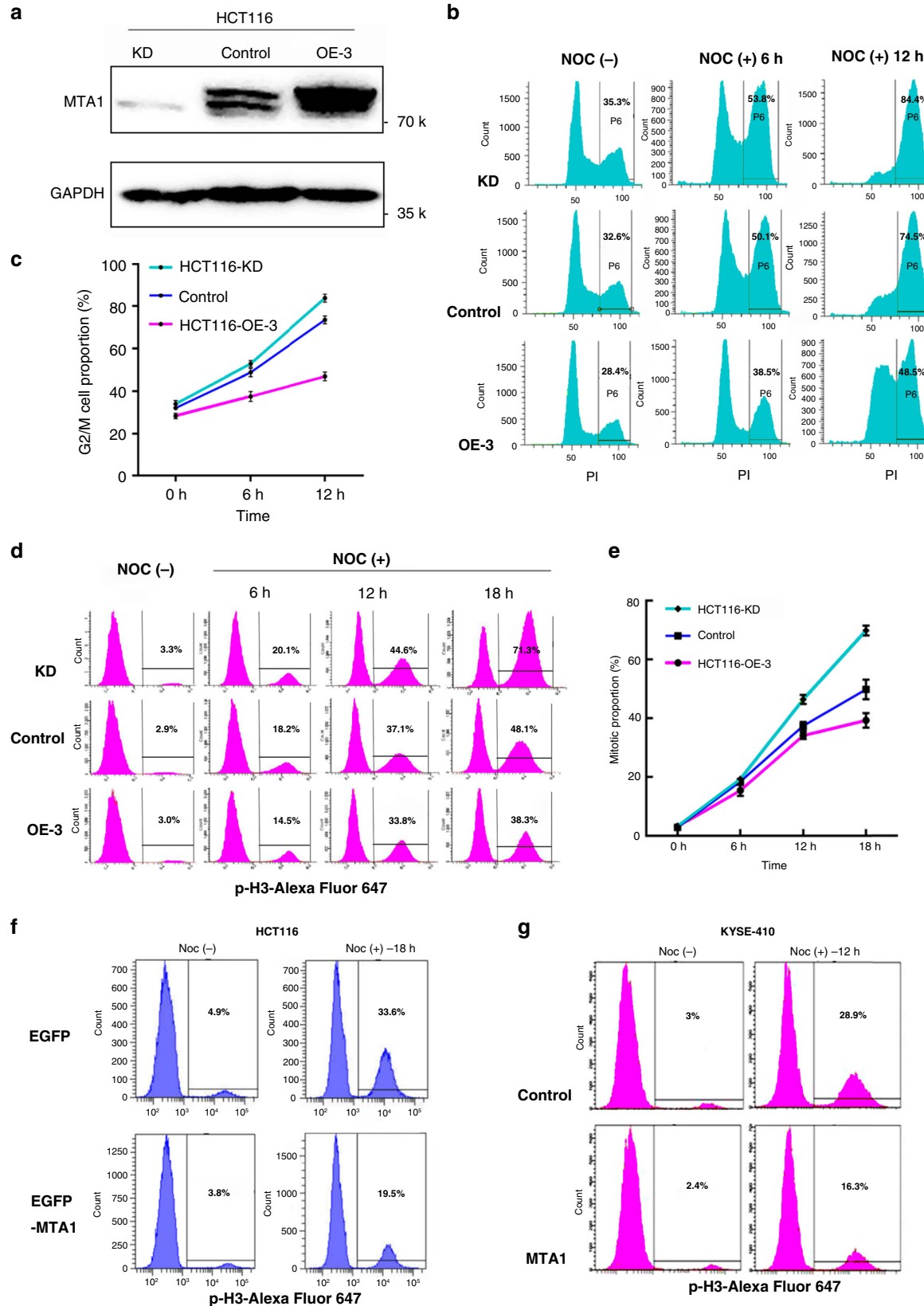

pulmonary metastases (2/6, 33.3%) in a nude mouse xenograft model, which were not found in the control mice (0/6, 0%) (Fig. 7g). In sphere formation assay, the MTA1-overexpressing cells formed more single cell-derived spheres with a larger, rounder and more compacted morphology through day 1 to day 8

compared to the control (~4.85-fold, $p < 0.01$, Fig. 7h, i, Supplementary Fig. 9h), suggesting MTA1 drives oncogenic growth. Moreover, the MTA1-overexpressing subclones displayed enhanced tumor initiation ability in the nude mice (Supplementary Table 2, Fig. 7j); as low as $1 \times 10^5$ MTA1-overexpressing cells

**Fig. 4 MTA1-transcript association regulates mitotic transition. a** Western blot verification of the knockdown (KD) or overexpression (OE) of MTA1 in the HCT116 cells. Shown is representative of three independent repeats with similar results. **b, c** Flow cytometry measurement of the mitotic-arrest cell proportion in MTA1-deletion, control or MTA1-overexpression HCT116 cells after nocodazole treatment for 6 h and 12 h. Propidium Iodide (PI) was used to stain the nuclear DNA, and the G2/M cells were marked with 4 N DNA content. **d, e** Flow cytometry measurement of the mitotic-arrest cell proportion after the nocodazole treatment for 6 h, 12 h and 18 h. The mitotic cells were marked by antibody against mitosis-specific phosphorylated Histone 3. **f** Flow cytometry measurement of the mitotic-arrest cell proportion in control and EGFP-MTA1 stably expressing HCT116 cells after nocodazole treatment for 18 h. **g** The overexpression of MTA1 in the KYSE-410 cancer cells also resulted in a lower mitotic arrest by nocodazole. **c, e** Data are represented as mean ± SD, $n = 3$ per group.

could initiate tumors, while the controls failed, indicating that MTA1 promotes oncogenesis in vivo (Supplementary Table 2, Fig. 7j).

To generally describe the MTA1-RNA association during oncogenesis, we plotted the MTA1-RBP correlation in tumor (TCGA pan-cancer), adjacent (TCGA pan-adjacent) and normal (GTEx pan-normal) tissues. We found the tightly correlated MTA1-RBPs coexpression was significantly loosened during tumorigenesis. We illustrated the correlations of MTA1 with thirty reported splicing factors, with MTA1-GAPDH correlation as an irrelevant control (Supplementary Fig. 9i, j), supporting that MTA1-RNA association-related posttranscriptional regulation is linked to tumor initiation.

## Discussion
RNA transcription and processing are both traditionally thought to be silenced in open mitosis when transcriptional machinery is largely expelled from chromatin[44,45]. The existence of transcription during mitosis is gradually accepted now[46–48], but the RNA splicing during mitosis is still seriously ignored. A recent sequential cell cycle transcriptome analysis revealed cell cycle-dependent splicing patterns involving approximately 1300 genes, and indicated the mitosis regulator *AURKB* with a mitosis-specific AS pattern[4]. Grenfell et al[5]. showed that mitotic RNA processing promoted kinetochore and spindle assembly in *Xenopus* and RNA splicing inhibition led to spindle defects. Our data also detected a spectrum of mitotic mRNA processing events, and suggested a delicate AS pattern switch of mitosis regulators, such as *ATRX* and *MYBL1*, during mitosis.

Recent advances in genome-wide studies have blurred the boundaries between chromatin-related activities and post-transcriptional RNA processes[49–52]. RNA-binding CAPs with both DNA- and RNA-binding activities may contribute to orchestrating these processes. Though have been recognized for years, less have been disclosed for RNA-binding CAPs on the role and mechanism underlying CAP-RNA interactions[53–55]. Recent improvements in protein-RNA crosslinking technologies have defined more and more multifunctional proteins with dual nucleic acid specificities, of which many are typical CAPs without classical RNA-binding domains (RBDs)[18,53,55,56]. Here, we show using a modified fCLIP-seq method that, MTA1, a known chromatin-modifying CAP, also contacts bulk mRNAs and regulates their processing.

MTA1-interacting proteins identified by co-IP-MS highlights a global MTA1 function in both DNA- and RNA-related processes, which was supported by *MTA1*-coexpressed genes in TCGA, CCLE and PDX samples. Besides the NuRD components, RBPs constitute the majority of MTA1 potential interactors in the co-IP-MS data. As lack of IgG controls, which may cause a higher false positive rate in MTA1 interactor screening, we compared the MTA1 potential interactome with others obtained from public resources including CRAPome, which showed that MTA1 antibodies captures a much higher proportion of RBPs than IgGs and non-RBP antibodies, but in a comparable level with RBP antibodies (Supplementary Table 3).

We also found MTA1-RBP and MTA1-RNA interaction clues from previous reports. HnRNPQ (SYNCRIP), a potential MTA1 interactor in our data, captured MTA1 (one of 105 total prey proteins) in a proteome-wide quantitative BAC-GFP inter-actomics survey[57]. Several prey proteins of HnRNPQ, such as YBX1, HNRNPAB, EIF2B1 and TUBB3[57], are also targets of MTA1 in our study. Also in this report, MTA1 baited 13 NuRD components and 6 RBPs (35 total prey proteins)[57], supporting MTA1's interaction with RBPs. He et al.[55] recently mapped protein-RNA interactions in mouse embryonic stem cells and identified a total of 803 proteins, including MTA1, with RNA binding activities. MTA1 is also a candidate of cell cycle-controlling lncRNA *ZFAS1*-binding proteins[58]. All these clues support MTA1 as an RNA-binding CAP.

Distribution of MTA1 and its association with chromatin in the cell are cell-cycle dependent[10]. At interphase, MTA1 localizes mainly to the nucleus, yet with a little fraction in the cytoplasm[8,9]. Upon entering mitosis, nuclear MTA1 disassociates from the chromatin at the start of prophase, segregates from the chromosome at metaphase and anaphase, and re-enters the nucleus when the nuclear membrane forms at telophase[10]. Since MTA1 is free of chromosome during most mitotic stage, the MTA1-induced mitosis-specific transcriptome alterations are probably irrelevant to transcription but to posttranscriptional regulation. This is consistent with our finding that the RNA-binding rather than the promoter-binding activity of MTA1 correlates with mitotic transcriptome alteration.

Despite decades of intensive efforts in RBP repertoire identification, our understanding of protein-RNA interactions still conceptually relies on the canonical RBDs. However, recent systematic profiling of nucleic acid–protein interactions have introduced numerous RBPs without classical RBDs. The expanding RBP repertoire contains a number of CAPs, like EZH2, SUZ12, CTCF, TET2 and P53[53,55]. Roles of CAP-RNA interaction in cell biology are being realized but are largely overlooked. Recent evidences indicate that CAP-RNA interaction regulates chromatin modifications and dynamics by guiding the chromatin modifiers to specific genomic locations[59–61]. In addition, the CAP-RNA interaction mechanism may be employed to modulate posttranscriptional RNA processes. The DNA-binding transcription factor p53 has been understood to possess post-transcriptional RNA binding activity for years[62]. Brm in the chromatin remodeling SWI/SNF complex regulates pre-mRNA processing through a RNA-binding mechanism independent of its DNA-binding activity[63–65]. Here, we show that MTA1-RNA association is more prevalent than DNA-binding in altering mRNA abundance and alternative splicing.

Though RNA processing was traditionally thought to cease during mitosis[3], our data indicate that MTA1, as an RNA-binding CAP, directs mitosis-specific AS pattern switches of many mitosis regulators during mitosis. By modulating the transcript abundance and AS pattern of mitosis regulators, MTA1 forces mitotic transition even in the presence of mitotic defects, leading to CIN and eventually tumorigenesis. Indeed, the MTA1-overexpressed cancer cells formed more cancer stem cell spheres in vitro and initiated more tumors in mice. Moreover, a higher frequency of

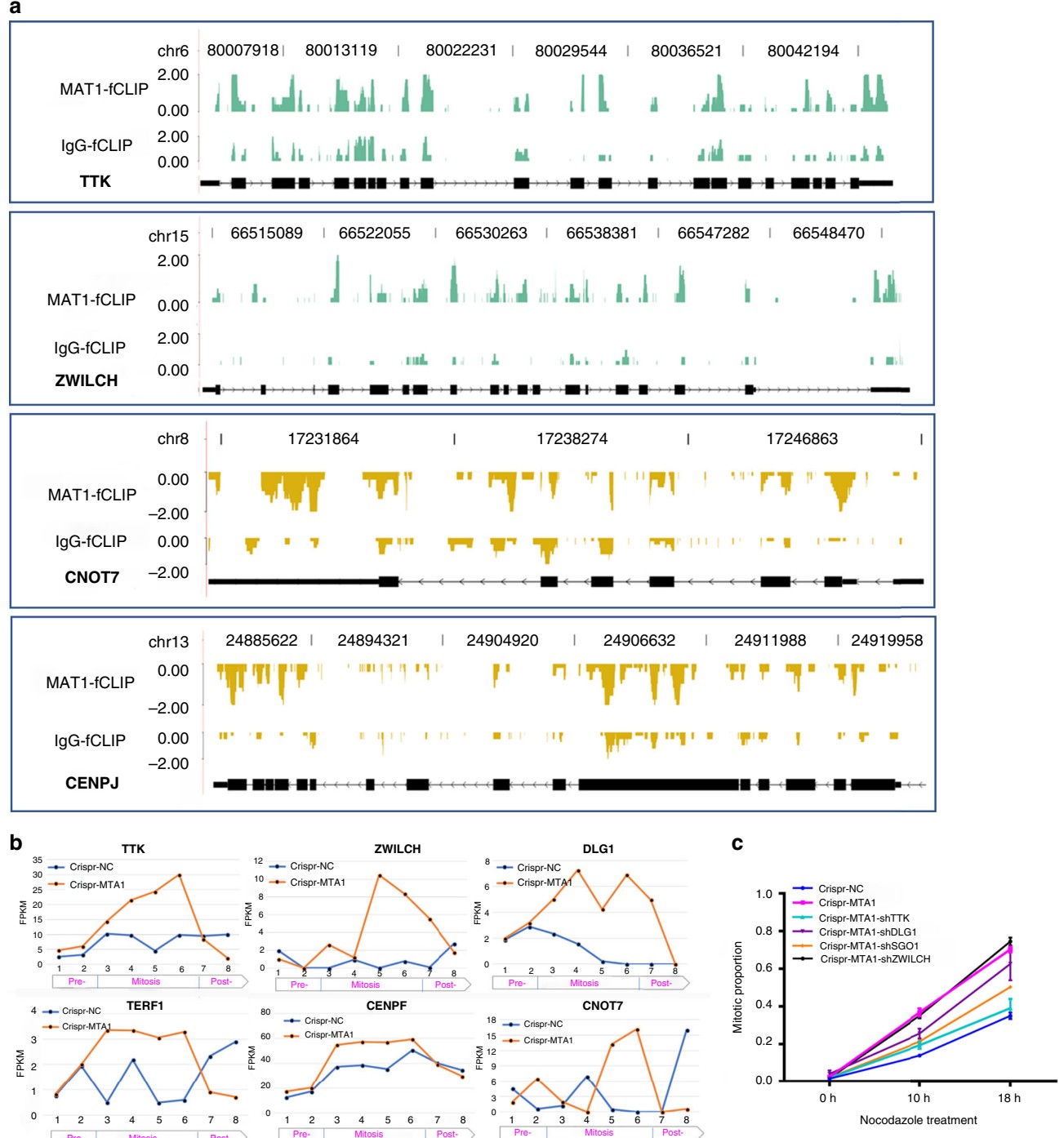

**Fig. 5 MTA1 alters transcript abundance of mitosis regulators. a** fCLIP-seq read density illustration showing the specific binding of MTA1 to transcripts of mitosis regulators, such as TTK, ZWILCH, CNOT7 and CENPJ. **b** MTA1 deletion led to a significant alteration of MTA1-associated mitosis-regulator transcripts in mRNA abundance during mitosis. For TTK, ZWILCH, DLG1 TERF1 and CENPF, the alteration started specifically after entry into the mitotic stage. While for CONT7, MTA1 deletion reversed the periodic fluctuation trends across the cell cycle. **c** TTK, ZWILCH, SOG1 and DLG1 were knocked down in MTA1-knockout HCT116 cells, and then treated with nocodazole for 0 h, 10 h and 18 h. The mitotic cells were marked with p-H3 antibody and counted using high content analysis system. Data are represented as mean ± SD, n = 3.

mitotic defectiveness was detected in colon cancer samples with elevated MTA1 expression.

In summary, we demonstrate that the chromatin modifier MTA1 interacts with abundant RBPs, contacts bulk mRNAs and preferentially targets mitosis-specific mRNA processing to drive mitotic transition, CIN and tumorigenesis.

## Methods

**Co-IP and mass spectrometry (MS)**. The HCT116 cells were lysed in NP-40 lysis buffer (50 mM Tris-HCl, pH 8.0, 0.5% NP-40, 10% Glycerol, 150 mM NaCl, 2 mM MgCl2, and 1 mM EDTA, supplemented with protease inhibitors). A total amount of 4 mg of HCT116 cell lysate was incubated with 12 μg of the primary mouse monoclonal MTA1 antibody (Abcam) or 3 μg of the rabbit polyclonal MTA1 antibody (Abcam) overnight at 4 °C on a rotary vibrator followed by incubation

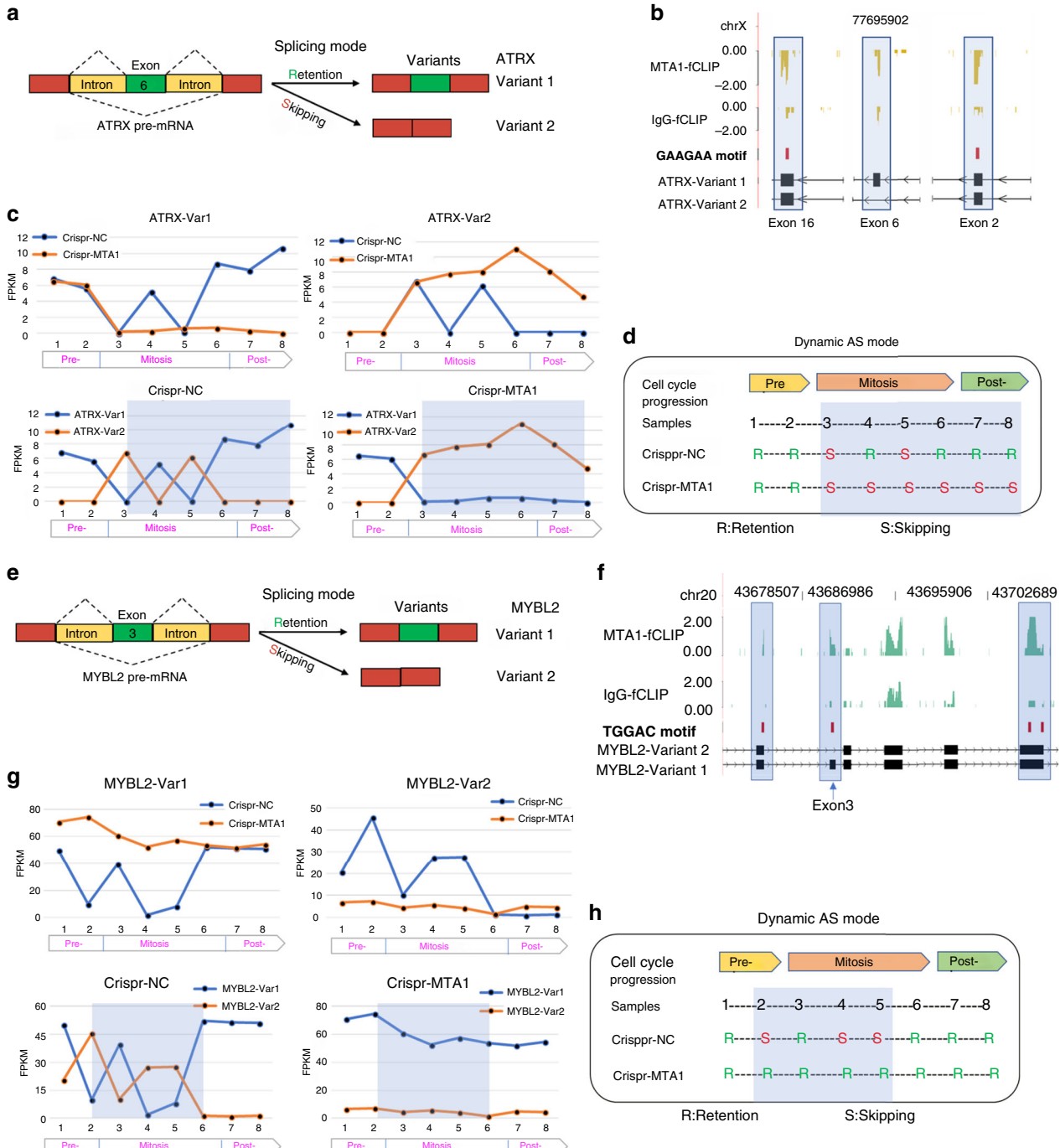

**Fig. 6 MTA1 drives mitosis-specific AS switch of mitosis regulators. a** The alternative splicing mode of the *ATRX* pre-mRNA. Variant 1 was produced with exon 6 retention, while variant 2 was with exon 6 skipping. **b** fCLIP-seq read density plot showing the specific binding of MTA1 to exon 6 and the GAAGAA motifs. **c** MTA1 regulates both the AS pattern and the AS dynamics of *ATRX* during mitosis. Var1, variant 1. Var2, variant 2. **d** The dynamic changes in the AS mode of *ATRX* with the cell cycle progression. R: exon 6 retention; S: exon 6 skipping. **e** The alternative splicing mode of the *MYBL2* pre-mRNA. Variant 1 was produced with exon 3 retention, while variant 2 was with exon 3 skipping. **f** fCLIP-seq read density plot showing the specific binding of MTA1 to exon 3 and the TGGAC motifs. **g** MTA1 regulates both the AS pattern and the AS dynamics of *MYBL2* during mitosis. **h** The dynamic changes in the AS mode of *MYBL2* with the cell cycle progression. R: exon 6 retention; S: exon 6 skipping.

with 80 μl of protein A/G beads (Santa Cruz) for 5 h at 4 °C. The immunoprecipitates were washed 6 times with NP-40 buffer and were collected after centrifugation at 3500*g* for 5 min. Subsequently, the beads were boiled in SDS loading buffer for 10 min, and the supernatant was concentrated with Amicon Ultra 0.5 ml 10 K centrifugal filter (Millipore) before SDS-PAGE separation. The gels were stained with Coomassie brilliant blue (Invitrogen) after electrophoresis. Each gel lane was separated into 5 sections, which were processed to mass spectrometry identification independently at Shanghai Applied Protein Technology Co., Ltd. The gel pieces were destained with 30% ACN/100 mM NH₄HCO₃. The gels were dried

in a vacuum centrifuge. The in-gel proteins were reduced with dithiothreitol (10 mM DTT/100 mM NH₄HCO₃) for 30 min at 56 °C and were then alkylated with iodoacetamide (200 mM IAA/100 mM NH₄HCO₃) in the dark at room temperature for 30 min. The gel pieces were briefly rinsed with 100 mM NH₄HCO₃ and ACN, respectively. The gel pieces were digested overnight in 12.5 ng/μl trypsin in 25 mM NH₄HCO₃. The peptides were extracted three times with 60% ACN/ 0.1% TFA. The extracts were pooled and dried completely by a vacuum centrifuge. The EttanTM MDLC system (GE Healthcare) was applied for the desalting and separation of the tryptic peptides mixtures. In this system, the samples were

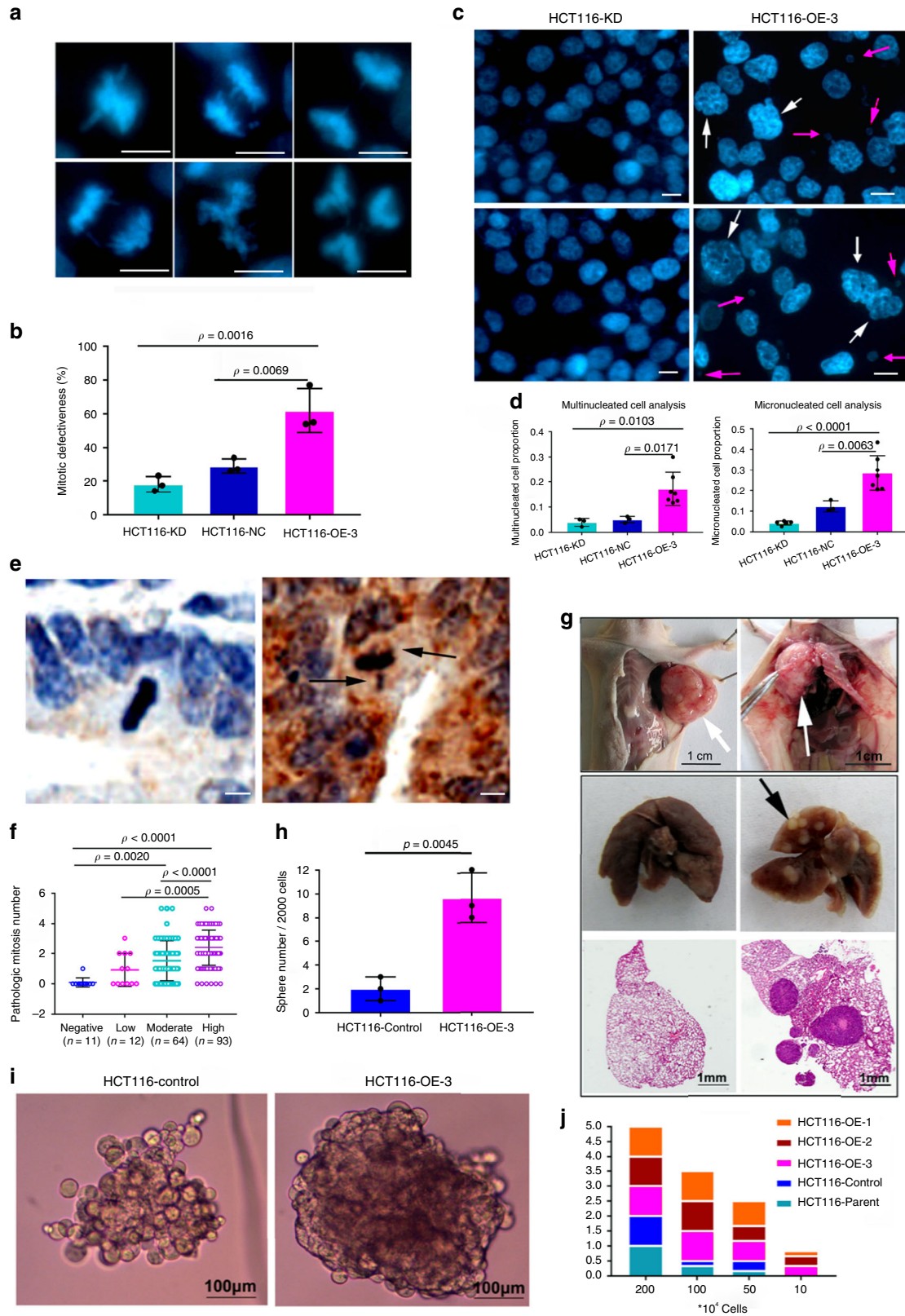

desalted on RP trap columns (Zorbax 300 SB C18, Agilent Technologies) and were then separated on an RP column (150 µm i.d., 100 mm length, Column technology Inc., Fremont, CA). Mobile phase A (0.1% formic acid in HPLC-grade water) and the mobile phase B (0.1% formic acid in acetonitrile) were selected. A total of 20 µg of the tryptic peptide mixtures was loaded onto the columns, and separation was done at a flow rate of 2 µl/min by using a linear gradient of 4–50% buffer B for 50 min, 50–100% buffer B for 4 min and 100% buffer B for 6 min. The LTQ Velos (Thermo Scientific), equipped with a microspray interface, was connected to the

LC setup for the detection of the eluted peptides. Data-dependent MS/MS spectra were obtained simultaneously. Each scan cycle consisted of one full scan mass spectrum (m/z 300-1800) followed by 20 MS/MS events of the most intense ions with the following dynamic exclusion settings: repeat count 2; repeat duration 30 s; and exclusion duration 90 s. The MS/MS spectra were automatically searched against the ipi.HUMAN.v3.53 using Bioworks Browser rev. 3.1 (Thermo Electron, San Jose, CA.). The protein identification results were extracted from SEQUEST out files with BuildSummary. The peptides were constrained to be tryptic and up to

**Fig. 7 MTA1 overexpression leads to CIN and tumorigenesis. a, b** Microscopic detection of mitotic defectiveness in HCT116 cell lines with MTA1 deletion or overexpression ($n = 3$ per group). MTA1 overexpression generated more mitotic defectiveness, including chromosome lagging, bridging, misalignment and multipolarized chromosomes. **c, d** Microscopic visualization of chromosomal instability in HCT116 cell lines with MTA1 deletion or overexpression. MTA1 overexpression resulted in a significantly higher occurrence of both multinucleated (**c** white arrow) and micronucleated (**c**; pink arrow) HCT116 cells. For multinucleated cell analysis, $n = 3$ except OE-3 in which $n = 7$; For micronucleated cell analysis, KD ($n = 5$), NC ($n = 3$) and OE-3 ($n = 7$). **e** Representative IHC images showing normal mitosis in MTA1-low colon cancer tissue (left) and pathological mitosis (black arrow) in MTA1-high colon cancer tissue (right). **f** The number of pathological mitosis cells in cancer tissues with negative, low, moderate or high MTA1 level. **g** Obvious pleura invasion (white arrow) and pulmonary metastases (black arrow) were found in the mice injected with the MTA1-overexpressing HCT116 cells but not in the control group. **h** The MTA1 overexpressing cells formed more single cell-derived tumor stem cell-like spheres than the control ($n = 3$ per group). **i** Two representative images showed that the spheres formed from the MTA1-overexpressing HCT116 cells were morphologically larger, rounder and tighter than those from the control cells. **j** Cumulative bar diagram showing the tumor occurrence rate of each group. All the cell lines initiated 100% tumors when injected with $2*10^6$ cells, while when injected $1 \times 10^5$ cells, only the MTA1-overexpressing cell lines initiated tumors. **b, d** and **f** One-way ANOVA with Tukey's multiple comparisons test. **h**, two-tailed Student's t test. All error bars represent mean ± SD. Results in **a, c, e, g** and **i** are representative of three independent repeats with similar results. Scale bar = 10 μm for **a, c** and **e**.

two missed cleavages were allowed. Carbamidomethylation of the cysteines was treated as a fixed modification, whereas oxidation of the methionine residues was considered as the variable modification. The mass tolerance allowed for the precursor ions was 2.0 Da, and the fragment ions were 0.8 Da, respectively. The protein identification criteria were based on the Delta CN ($\geq 0.1$) and cross-correlation scores (Xcorr, one charge $\geq 1.9$, two charges $\geq 2.2$, three charges $\geq 3.75$).

For the co-IP identification of the interaction, 500 μg to 1 mg of the HCT116 cell lysate, 1–2 μg of the primary antibody and 20 μl of the protein A/G were used.

**CRISPR-cas9 mediated *MTA1* gene-editing.** To knock out MTA1 in the human HCT116 colon cancer cell line, we designed two pairs of single guide RNA (sgRNA) sequences for human *MTA1* using the design tool from the Feng Zhang Lab and cloned the targeting sequences into the lentiCRISPRv2 vector (a gift from Feng Zhang; Addgene plasmid #52961). The lentiviruses for the *MTA1* sgRNAs or the vector control were generated in the HEK293T cells by the cotransfection of LentiCRISPRv2 with the packaging vectors pMD2G and psPAX2 using lipofectamine 2000 (Invitrogen). The HCT116 cells were infected with lentivirus for 72 h and were then selected with 2 μg/ml puromycin. The MTA1 protein level was assessed using a Western Blot two weeks after the puromycin selection. The target sgRNA sequences are as follows:

sgRNA-*MTA1*-1 5′-CACCGCTCTGCCCGCCACGCACATC-3′ 5′-AAACGA TGTGCGTGGCGGGCAGAGC-3′ sgRNA-*MTA1*-2 5′-CACCGATCGGGAGCT GTTCCTCTCC-3′ 5′-AAACGGAGAGGAACAGCTCCCGATC-3′

**Immunofluorescence and in situ PLA.** The co-localization of MTA1 with SMN, RPS3, YBX1 and PTBP1 was carried out using immunofluorescence detection. Briefly, the cells were grown on sterilized coverslips and were fixed by 4% paraformaldehyde at room temperature for 15 min, washed twice in PBS, permeabilized with 0.25% Triton X-100 at room temperature for 10 min and blocked with 0.5% bovine serum albumin for 30 min. The coverslips were subsequently incubated overnight with the primary mouse monoclonal MTA1 antibody and rabbit antibodies against SMN, RPS3 or YBX1 followed by a 1 h incubation in the corresponding fluorescent secondary antibodies diluted in blocking buffer. The coverslips were finally mounted with mounting medium containing DAPI. The images were acquired using fluorescence (Olympus) or confocal laser scanning (Leica) microscopy.

The in situ PLA experiments were performed to visualize the in situ interaction of MTA1 and SMN in the HCT116 cells using the Duolink kit (Olink Biosciences AB) and were performed according to the Duolink kit protocol. The images were acquired using fluorescence microscopy (Olympus).

**Cell culture and drug treatment.** The HCT116 and HEK293T cell lines were obtained from the National Infrastructure of Cell Line Resources (Beijing, China). KYSE410 was a kind gift from Dr. Takayoshi Tobe at Kyoto University. All the cell lines used in this study were cultured in DMEM medium (Gibco) supplemented with 10% fetal bovine serum (FBS) (HyClone) and 1% penicillin/streptomycin (HyClone) at 37 °C in a humidified incubator with 5% $CO_2$. For the mitotic arrest induction, 100 ng/ml nocodazole was used. For cell synchronization at the G1/S phase, 2 mM thymidine was used.

**RNA-seq library construction and data analysis.** The total RNA was treated with RQ1 DNase (Promega) to remove the DNA. The quality and quantity of the purified RNA were determined by measuring the absorbance at 260 nm/280 nm (A260/A280) using smartspec plus (Bio-Rad). RNA integrity was further verified by 1.5% agarose gel electrophoresis. For each sample, 5 μg of total RNA was used for RNA-seq library preparation.

The polyadenylated mRNAs were purified and concentrated with oligo (dT)-conjugated magnetic beads (Invitrogen) before being used for the directional RNA-seq library preparation. The purified mRNAs were ion fragmented at 95 °C followed by end repair and 5′ adaptor ligation. Then, reverse transcription was performed with RT primers harboring the 3′ adaptor sequence and a randomized hexamer. The cDNAs were purified and amplified, and the PCR products, corresponding to 300–500 bps, were purified, quantified and stored at −80 °C until they were used for sequencing.

For non-synchronized cell RNA-seq, the libraries were sequenced on an Illumina Next500 for 150-nt pair-end reads, following the manufacturer's instructions, by ABlife Inc. (Wuhan, China). For synchronized sample RNA-seq, the libraries were sequenced on a HiSeq X for 150-nt pair-end reads.

After obtaining the raw reads from the sequencing platform, we used cutadapt (version 1.7.1) and FASTX Toolkit (Version 0.0.13) to remove adapters and low quality bases (30% bases quality less than 20). The clean reads were mapped to the human genome (GRCH38) by TopHat2 allowing no more than 4 mismatches. The uniquely mapped reads were obtained to calculate the read number and FPKM (fragments per kilobase and per million) values for each gene.

We used edgeR software to analyze the differentially expressed genes (DEGs). We compared the crMTA1 samples and control samples to investigate the genes that were regulated by MTA1. For each gene, the p-value was obtained based on the model of the negative binomial distribution. To define the DEGs, a 0.01 FDR (false discovery rate) was set as the threshold.

**Alternative splicing analysis.** The alternative splicing events (ASEs) and regulated alternative splicing events (RASEs) between non-synchronized controls and crMTA1 samples were defined and quantified by using the ABLas pipeline[21]. Briefly, the detection of seven types of ASEs was based on the splice junction reads. The eight types of ASEs included Cassette exon (CassetteExon), Exon skipping (ES), Mutual exclusive exon skipping (MXE), A5SS, A3SS, the MXE combined with alternative 5′ promoter (5pMXE), and with alternative polyadenylation site (3pMXE). Intron retention was calculated by the reduced usage of the candidate splice sites. ABLas defined the intron retention (IR) splicing events, with the following criteria: (1) the boundary reads were at either the 5′ or 3′ splice site of the candidate; (2) the mean base depth in the candidate intron was at least 20% of the flanking exon and twice that of the intronic depth in the model gene. The ratio of the mean base depth in the candidate intron to the flanking exon was considered the IR intensity.

After detecting the ASEs in each RNA-seq sample, a Fisher's exact test was used to calculate the significant p-values, with the alternative reads and model reads of samples as input data, respectively. The changed ratio of the alternatively spliced reads and the constitutively spliced reads between the compared samples was defined as the RASE ratio. The p-value < 0.05 and RASE ratio > 0.2 were set as the thresholds for the detection of RASEs.

For synchronized sample isoforms detection, we used rmats 4.0.1 for quantification and annotation.

**qRT-PCR validation of ASEs.** For each of the AS events, there are two different types of spliced isoforms, including the model (M) and the alternative form (A). Two to three pairs of primers were designed to amplify the model or alternative forms of the spliced products.

The experimental design for the IR validation is diagrammed below (Supplementary Fig. 10a). The reverse primer for the model isoform spanned the spliced junction, while that for the alternative isoform permitted the amplification across the exon-intron boundary.

The primer design for the Cassette Exon events is shown in Supplementary Fig. 10b. The model isoform skips the alternative exon (exon 2), while the alternative isoform includes the alternative exon 2. The primers for the other ASEs were designed similarly.

For the qRT-PCR, approximately 1 μg of the total RNA was used for reverse transcription, and the cDNA was synthesized with M-MLV Reverse Transcriptase

(Vazyme). The real-time PCR was then performed using a QuantStudio 6 Flex System (ABI) according to the manufacturer's standard protocol. GAPDH was the endogenous control and was used to normalize the amount of each sample. The assays were repeated at least three times. All the values are expressed as the threshold crossings (Ct). The ratio was calculated according to As RQ/Model RQ. All the As-specific primers and Model-specific primers for the qRT-PCR are listed in Supplementary Table 4.

**fCLIP-seq library construction and data analysis**. Approximately $1 \times 10^7$ HCT116 cells were grown in Petri dishes and were treated. After the growth media was removed, cells were rinsed twice with cold $1 \times$ PBS. Formaldehyde was added to a final concentration of 1% followed by a gentle mix and an incubation at room temperature for 10 min. To stop the crosslinking reaction, glycine was added at a final concentration of 0.125 M. The cells were harvested in cold PBS by scraping and were transferred into a 1.5 ml microcentrifuge tube and were centrifugation at $1000 \times g$ for 5 min at 4 °C. The collected cells were lysed on ice by resuspending in and repeated pipetting with lysis buffer ($1 \times$ PBS, 0.1% SDS, 0.5% NP-40 and 0.5% sodium deoxycholate) (10 volumes of the cell pellets). Then, 1% (vol/vol) 40U/ul Recombinant RNase Inhibitor (Takara) and 2% (vol/vol) 10 mg/ml PMSF were added to the cell lysate followed by rotating the mixture end over end at 4 °C for 30 min. Then, DNase I was added (Promega) followed by an incubation at 37 °C for 3 min. Stop solution was then added to quench the DNase. Then, the samples were centrifuged at $9600g$ for 10 min at 4 °C. The supernatants were carefully collected. Then, the CLIP procedure was conducted, with the exception that the gel fractionation of the protein-RNA complex step was omitted. Then, a 1/50000 dilution of MNase (Thermo) was prepared to fragment the RNAs. Complementary DNA (cDNA) libraries were prepared with the Balancer NGS Library Preparation Kit for small microRNA (K02420, Quest Genomics) according the manufacturer's procedure. The libraries were sequenced on an Illumina Next500 for pair-end reads, following the manufacturer's instructions, by ABlife Inc. (Wuhan, China).

The CLIP-seq clean reads were also aligned to the human genome by TopHat2[19] allowing 2 mismatches. After that, we removed multiple mapped reads and duplicated the reads and used the ABLIRC strategy[21] to identify the MTA1 binding peaks. We used Homer software[20] to call the MTA1 binding motifs.

To calculate the frequencies of the 5'ss and 3'ss motifs in the peaks located in the CDS region, the 5'UTR region, the 3'UTR region, the intron region, the antisense strands, and the exon-intron boundaries, we used fuzznuc[66] to calculate the proportions of the motifs [GA]GT[AG]AG and T[TCG][TCG]N[TC]TN[TC]AG in the different regions and presented the results by the bar graph.

**ChIP-seq library construction and data analysis**. Formaldehyde crosslinked cells, prepared the same as for the fCLIP-seq, were lysed on ice by resuspending and repeated pipetting with 10 cell-pellet volumes of RIPA buffer (50 mM Tris 7.4, 150 mM NaCl, 2 mM EDTA, 0.1% SDS, 0.5% NP-40 and 0.5% deoxycholate). The cells lysates were sonicated to generate DNA fragments of 200–500 bp, were centrifuged for 10 min at 12,000 g, and the supernatant was directly used for immunoprecipitation. The ChIP experiments were performed as previously described[20]. Pierce™ ChIP-grade Protein A/G Magnetic Beads were used in the ChIP experiments. A total of 10 μg MTA1 antibody and Rabbit IgG were used. The DNA recovered from the beads were used to generate the ChIP-seq DNA libraries using the ThruPLEX® DNA-seq Kit (R400427, Rubicon Genomics) according the manufacturer's procedure. The libraries were sequenced on an Illumina Next500 for pair-end reads, following the manufacturer's instructions, by ABlife Inc. (Wuhan, China).

The clean reads were mapped to the genome by Bowtie2[56]. Uniquely aligned reads were used to identify the MTA1 binding sites by MACS14[67]. The input sample was treated as the control. All the peaks in each sample were clustered by bedtools[68] (peaks with at least 1 bp overlap were merged together). To construct the relationship between the peaks and genes, a window between 10 kb upstream and 3kb downstream of the transcription start site (TSS) was set to obtain the promoter binding peaks of the corresponding genes.

**In vitro tumorsphere formation assays**. For the tumorsphere formation assay, the cells were fully digested into single cells, and 2000 cells in 2 ml serum-free DMEM/F12 medium (Invitrogen) supplemented with 10 ng/ml basic fibroblast growth factor (Sigma), 20 ng/ml epidermal growth factor (Sigma), 5 μg/ml insulin (Sigma), $1 \times$ B27 supplement (Invitrogen) and 0.4% bovine serum albumin (Sigma) were plated into ultralow attachment 6-well plates (Corning). The cells were cultured under 5% $CO_2$ at 37 °C for 8 days, and the growth of the tumorspheres was monitored by microscopy every day.

**Xenograft Assays**. For the subcutaneous xenograft experiments, female NU/NU nude mice, at 5–6 weeks of age, were purchased from the Beijing Vital River Laboratory Animal Technology Co., Ltd. The experimental protocols performed on the animals were approved by The Institutional Animal Care and Use Committee of Cancer Hospital, Chinese Academy of Medical Sciences & Peking Union Medical College. For the metastasis analysis, $2 \times 10^6$ cells were subcutaneously injected, the mice were sacrificed and the lungs were collected for metastasis detection after one month. For the tumorigenesis analysis, $2 \times 10^6$, $1 \times 10^6$, $5 \times 10^5$ or $1 \times 10^5$ cells were injected subcutaneously. All the mice were monitored every

day for tumor development. The mice were humanely euthanized 2 to 12 weeks after implantation, when the tumor size reached ~1500 mm³, and the tumors were collected for further pathology confirmation.

**Clinical colon cancer specimens**. The present study was approved by the Ethics Committee of Cancer Hospital, Chinese Academy of Medical Sciences & Peking Union Medical College. Human colon cancer and the adjacent specimens were collected from 2012.08 to 2016.07 at the Cancer Hospital, Chinese Academy of Medical Sciences & Peking Union Medical College. These clinical colon cancer specimens were examined and diagnosed by pathologists at the Cancer Hospital, Chinese Academy of Medical Sciences & Peking Union Medical College. All the human samples were obtained after obtaining informed consent from the patient.

**Other statistical analysis**. To analyze the functional biology of the genes, the DAVID Bioinformatics Resources 6.8 (https://david.ncifcrf.gov/), GeneOntology (http://www.geneontology.org/) and KEGG (Kyoto Encyclopedia of Genes and Genomes)[69] databases were used for the enrichment analysis, and a Fisher's Exact Test was used to define the enrichment degree. Heatmap was generated by pheatmap, which is one of the R packages. The network was generated by Cytoscape[70]. The other Figures were generated by the native R program and the Python program. Unless otherwise stated, statistical analyses were performed using R program or GraphPad Prism (v7.0) and $p$-value < 0.05 was considered as statistically significant.

**Reporting summary**. Further information on research design is available in the Nature Research Reporting Summary linked to this article.

## Data availability
The fCLIP-seq, ChIP-seq, non-synchronized and synchronized RNA-seq data reported here are available under GEO Super Series GSE123959. The mass spectrometry proteomics data have been deposited into the ProteomeXchange Consortium via the iProX partner repository with the dataset identifier PXD018242. The source data underlying Figs. 1a, d, e, 2b, d, h, i, 3b, h, j, 4a, c, e, 5b, c, 6c, g, 7b, d, f, h, j and Supplementary Figs. 1a–e, g, n, 2c–e, 3a, b, e, f, h, j, 4a–c, 5a–b, 6, 7a, b, d, e, 8a–e, g, h, j, 9d, f, g, i are provided as a Source Data file. All the other data supporting the findings of this study are available within the article and its supplementary information files and from the corresponding author upon reasonable request. A reporting summary for this article is available as a Supplementary Information file. The public databases used in this study: Database for Annotation, Visualization and Integrated Discovery 6.8 (DAVID 6.8); Gene Ontology (GO, http://www.geneontology.org/); The Cancer Genome Atlas (TCGA); Gene Expression Omnibus (GEO); Cancer Cell Line Encyclopedia (CCLE); Encyclopedia of DNA Elements (ENCODE); The contaminant repository for affinity purification (CRAPome). Source data are provided with this paper.

## Code availability
The CLIP-seq peak calling program (ABLIRC) is publicly available at https://ablifedev.github.io/ABLIRC/. The RNA-seq AS program (ABLas) is publicly available at https://github.com/ablifedev/ABLas. Source data are provided with this paper.

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

## Acknowledgements

This work was financially supported by grants from the National Basic Research Program of China (973 Program) (No.2015CB553904), the CAMS Innovation Fund for Medical Sciences (CIFMS) (No.2016-I2M-1-001, 2017-I2M-3-004 and 2019-I2M-1-003), the National Natural Science Foundation of China (No. 81572842, 81672459, 81872280, 81502384, 81874122 and 81490753), the Non-profit Central Research Institute Fund of Chinese Academy of Medical Sciences (2017PT31029), the Open Issue of State Key Laboratory of Molecular Oncology (No. SKL-KF-2017-16), the Independent Issue of State Key Laboratory of Molecular Oncology (No. SKL-2017-16) and grant from ABLife (ABL2014-03005). We appreciate ABLife Inc., Wuhan for the production and analysis of fCLIP-seq and RNA-seq data.

## Author contributions

Conceptualization: H.Q., J.L., F.M., Q.Z. and T. Wen. Experiment: J.L., C.L., J.W., D.X., H.W., L.L., T. Wang., Y.W. and Y.Z. Data analysis: J.L, C.L., J.W., H.L., P.N., J.Z. and D.C. Resources: F.M., D.X., C.H., Q.Z., T. Wen. and H.Q. Manuscript writing: J.L. Manuscript editing: J.L., C.L., J.W. T. Wang., J.Z. and H.Q. Visualization: J.L., C.L., J.W., H.W., D.C. and Y.Z. Supervision: H.Q., F.M., Q.Z. and T. Wen. Project Administration: H.Q. and J.L. Funding Acquisition: H.Q., H.W., F.M. Q.Z. and J.L..

## Competing interests

The authors declare no competing interests.
