## [Peer Review File · Nature Communications]

Reviewers' comments:

Reviewer #1 (Remarks to the Author):

This paper reports a new role for MTA1 as a regulator of alternative splicing during mitosis. This finding has a high novelty factor as it reveals a hitherto unknown function of MTA1 and challenges the notion that mRNA processing is minimal during mitosis. Overall, these new claims are well supported by experimental data and extensive bioinformatic analysis. This paper is a prime example for the seamless integration of state of the art biological experimentation and computational analysis.

I only have some minor points of criticisms

1) the description of experiments and methods in the text and especially in the figure legends is often very sparse, which makes it difficult to follow experiments

2) What is not clear is the relationship between MTA1's ability to destabilise transcripts and to affect alternative splicing. This question is important as the authors show very profound effects of MTA1 knockout or overexpression on the abundance of mitosis related transcripts and alternative splicing. However, it is not clear whether the biological effects stem from the expression changes or from the alternative splicing. This question should be ideally addressed experimentally (although I concede that this may be difficult) or at least by discussion.

3) In the same vein, it would strengthen the authors main conclusions if clear roles for some of the MTA1 dependent alternatively spliced transcripts could be shown. For instance, how does the alternative splicing change the function of the resulting proteins?

4) Some more analysis of the MTA1 status in tumours also would be helpful for supporting the physiological significance of the paper. For instance, is there a correlation between MTA1 expression and mutations in splice factors?

5) Fig. 1E is insufficiently explained. It is unclear which antibodies were used for IP and which one for Western blotting.

6) Fig. 1G and S1M. At the resolution shown co-localisation is not apparent.

7) Fig. S7A. RANBP1 expression shows an opposite behaviour compared to the other transcripts. Any explanation for that?

8) Fig. S7C. The blot hardly shows any differences, while the bar graph claims significant differences. How often was this experiment repeated? How was statistical significance calculated?

Reviewer #2 (Remarks to the Author):

The manuscript by Liu et al. shows that MTA1, a protein that was previously characterized as a chromatin modifier, is also a RNA-binding protein that modulates splicing, in particular during mitosis. The authors start from an interactomics screen where they observed and confirmed the association with many RNA-binding proteins. Then an adapted CLIP-Seq protocol was performed to identify and characterize the binding sites on mRNA molecules pointing to the modulation of alternative splicing. The authors observed a strong link to mitosis and continued to show that splicing is affected by MTA1 during mitosis. Dysregulation of MTA1 leads to aberrant chromosome segregation and instability, and promotes tumorigenesis. By looking at clinical cancer specimens, a correlation was observed.

This is an extensive manuscript which comprehensively covers the RNA-binding aspect of MTA1.

The experiments appear well performed albeit somewhat difficult to follow because of limited information (see below). Apart from the concerns that I have expressed below, I think the data are of good quality and, as the manuscript touches upon an exciting field with many outstanding questions. Publication can be supported if the comments are sufficiently addressed.

Major comments:

While it does make sense to provide the data as a single story, the manuscript becomes really bulky and sometimes difficult to follow (see figure legends comment below). I assume that some data were left out to keep the manuscript streamlined (see f.i. my comment on the fCLIP-seq protocol). This inevitably raises questions on quality of the data (while there may not be an issue with quality). Therefore, the authors should consider making separate manuscripts.

As a general comment, it would help to provide some more information in the figure legends. The information is too limited now and figures are therefore hard to understand. Please also provide some information on the experiment that was performed to allow a good assessment of the data. Some abbreviations are also not explained. This will lead to a longer manuscript however.

While the authors provide an impressive amount of work, it would help if the initial observations relied on data that is somewhat more robust:

For the interactomics screening, the authors use two different anti-MTA1 antibodies to pull down the complexes associated to MTA1. Overlap is used as a criterion to assess quality. This is however not a valid approach to perform these experiments. Many proteins stick to the beadmatrix (or even the tube walls) and there is thus no way to make a distinction on what is specific and unspecific. More disturbingly is the fact that these sticky proteins are typically abundant proteins and many of them are binding RNA. Ribonucleoproteins belong to the most detected protein families in the CRAPome (Mellacheruvu, Nat Methods 2013) implying that there is a good chance to detect these proteins anyhow, independent of the protocol. A better approach would therefore be to include a number of control experiments in the design. Good controls are antibody controls (antibody pool from the same species but not specific for a protein target). For data analysis, either quantitative proteomics can be used (e.g. LFQ with MaxQuant and Perseus visualization), or tools such as SAINT, SFINX or others, which focus on spectral or peptide counts. The downstream validation supports (some of) the observations that were made using the ill-designed MS experiment, so it should be possible to confirm this. In addition, the mass spec data should be deposited in a repository (e.g. PRIDE).

The authors rely on an adapted protocol (fCLIP-Seq) to show the binding to RNA sites. While the data that was obtained may be true, it is risky to rely on a new protocol to make these statements. Are there other manuscripts in preparation or papers where this approach was described? It would definitely help to have some kind of positive control to show that this approach is actually performing as expected. This would help make sure that there are no unexpected artifacts that explain the observations.

Minor comments:

Line 67: rephrase

Line 105: Coupled with... rephrase this sentence

Line 133: 'is always indicative': this is a strong statement although it's still indicative. Best rephrase

Line 138: Typo in 'transcriptionally'

Line 396: what is exactly meant by this statement? Is this a reference to a specific protein-RNA interaction? Best rephrase

Line 401 and M&M linked to this: how were these KO cells generated exactly? Were cells infected with both gRNA constructs? If lentiviral expression remains in the infected cells, off-target sites will be cleaved also.

Line 423: ENCODE project reference incomplete

Line 429: How were the high, medium and low affinity CHIP targets defined? Total divided by 3 (implying equal numbers?)

Figure 3J does not show all results for the RT-qPCR experiments. How was the selection made on what is shown in the figure?

Line 530: rephrase 'we enriched'

Line 547: were single cell clones used in these experiments? If so: can clonal variation be a

problem? Also for Fig4F

Fig S7C: were these data normalized? How many times performed?

Figure 6 mostly relies on data extracted from the sequencing approaches. It would be good to confirm some of the observations using RT-qPCR.

Line 872: Would it make sense to include other members of the family to assess the specificity of this observation.

Line 894: I think 'Figure 7G' should be 'figure 7I'

Reviewer #3 (Remarks to the Author):

Liu and colleagues report a previously unknown role in alternative splicing regulation for the chromatin-associated protein MTA1.

First, by transcriptome analysis for several primary cancers, cell lines and patient derived xenografts authors show that genes correlated with MTA1 expression are involved in pre-mRNA processing and splicing, in addition to the predicted DNA repair and transcription functions. Direct mass spectrometry analysis confirmed that RNA-binding proteins are the major components of the MTA1 interactome. Next, sequencing of the RNAs co-precipitated with MTA1 following formaldehyde its interaction with several transcripts. Consensus sequence for the 5'splice-site represent the preferential binding motif for MTA1, whose binding is indeed enriched at the exon-intron boundary. RNA-seq analysis of MTA1 depleted cells highlighted a large number of genes and splicing events whose expression is controlled by MTA1. Notably, RNAs of both transcriptionally- and splicing-regulated genes are bound by MTA1, while poor binding is found by chip-seq on their promoters, suggesting that MTA1 influences RNA processing. They also suggest that MTA1 splicing activity regulates mitotic functions in cancer cells. In line with its role in mitotic regulation of cellular transcriptome, MTA1 ablation causes severe aneuploidy and mitotic defects in cancer cell lines; moreover, MTA1 expression levels correlates with the mitotic score of primary tumors and with the tumorigenicity of cancer cells, as shown by in vivo experiments using xenografts models in nude mice.

This manuscript deals with very interesting issues, such as post-transcriptional regulation by chromatin associated proteins and coupling between transcriptome-regulation and cell-cycle control in cancer cells.

However, many of statements and conclusions are not supported by direct experiments, but rather by correlation. Some experiments lack appropriate controls and in some key aspects the study is too descriptive (i.e. lack of exhaustive validation, absence of investigation of the mechanisms by which MTA1 exerts its splicing regulation). The authors suggest a role of MTA1 in transcript destabilization, but this is not investigated in the manuscript. Is it related to splicing or independent of it? MTA1 expression correlates with that of many splicing factors and spliceosome components, some of which interact with it. Is MTA1 required for their function? Why specifically in mitosis? Many splicing factors are also associated with chromatin and released in mitosis when chromatin condense like shown for MTA1. Are these events linked? Does MTA1 modulate the recruitment of the spliceosome or other splicing factors to the target genes? In general, how would MTA1 affect splicing? they should use few model genes to investigate the mechanism of action. More specific comments are listed below.

Major points

Figure 1. Western blot shown for co-ip experiment are of poor quality and do not show reliable interaction for many of the analyzed protein. For instance, the same amount of nono protein is found in the supernatant of mta1 and igG ip, suggesting that it has not been depleted by the co-ip. Same criticisms for co-ip treated with RNase/DNase in fig s1n, where many RBPs do not seem to associate with MTA1 in the absence of RNase. How do the authors explain this? Overall, such interactions appear extremely weak. Likewise, the immunofluorescence analysis of the association of MTA1 with actin and tubulin cytoskeleton do not appear to be conclusive, because there is no high resolution in the images shown and partial overlap is expected from the widespread

distribution of cytoskeletal proteins.

Figure 2. Authors state that they combined the advantages of formaldehyde-rip and clip technique. However, the greatest advantage of using clip technique is the chance to map only direct interaction between RNA and proteins crosslinked by the UV irradiation. This advantage is not exploited by authors. By performing formaldehyde crosslinking authors allowed retrieval of both direct and indirect interactions, as in a canonical rip-seq experiment. Could the author clarify what is the advantage? Given that MTA1 interacts with multiple DNA- and RNA-binding proteins, promiscuity between direct and indirect binding is very likely when a potent crosslinker like formaldehyde is used.

Fig2b. To test if mta1 binding is enriched in a peculiar genic region, authors compared results of their clip-experiment to RNA-seq reads. It is unclear which kind of sample this latter refers to. Nevertheless, a more appropriate comparison should have been the RNA-seq of the input RNA used for the immunoprecipitation experiment. A similar consideration holds also for fig2d: what is the background considered for the analysis of MTA1 motif localization?

Figure 3. Only 13 splicing-events are shown and only 8 are significant. Validation should be improved and regulated exons and/or introns should be clearly indicated in the figure.

Figure 4. How was stable overexpression and knockdown of MTA1 achieved? Why MTA1 appears as three bands in the western blot? What is the effect of MTA1 overexpression/downregulation on the cell cycle of cells without imposing a microtubule stress with nocodazole? Another interpretation of the data is that overexpression of MTA1 slows down the cell cycle, thus reducing the percentage of cells accumulating in G2/M in the time-frame of nocodazole treatments. Also, why different aspects of the mitotic defects correlated to variations of MTA1 expression levels are investigated using different models? Do crispr cas9 knockout cells show the same "mitotic defective" phenotype?

Figure s5. Authors state that MTA1 binding induces destabilization of bound transcripts. However, no data on stabilization of transcripts is shown here. What the data show is a change in expression levels, which may be caused by several mechanisms, including indirect transcriptional mechanisms (for instance related to cell cycle progression effects) not requiring direct binding of MTA1 to the promoter region of the regulated genes. Also, the authors refer to these genes as "mitotic transcriptome". What is the overlap with the MTA1 DEG genes described in the previous figure? Are these different genes or they mainly overlap with what described before? In general, the authors conclude this paragraph stating that MTA1 controls the mitotic transcriptome by an RNA-binding mediated mechanism. However, this statement is not supported by data, as the results are only correlative.

Figure 5 and s7. Authors correlate the mitotic defect observed in MTA1 depleted cells with the altered oscillatory expression of some mitotic regulators, whose transcripts are bound by MTA1. However, for none of these genes an independent validation has been carried out. Validation would corroborate this observation.

Figure 6 and s8. The same criticism for the lack of validation is raised for the MTA1- splicing regulated mitotic genes. Moreover, splicing regulation seems independent from MTA1 binding, because exon 16 and exon 2 of the ATRX gene show MTA1 binding and presence of its consensus motif, but their splicing is not regulated. Also, as the splicing of MYB and ATRX fluctuates in mitosis in control cells, how is MTA1 eventually regulated to be switched on and off so quickly?

Figure 7. Comparison toward samples with basal level of MTA1 should be performed in order to appreciate the mitotic phenotype variations induced by modulation of MTA1 expression levels. Statistical analysis should corroborate the significance of the difference in tumorigenicity observed in the model (figure7g).

Minor points

Figure legend of both figure 1 and s1 should be more detailed, describing for instance the cell line used for the illustrated experiment.

Figure s6. In this figure non-mitotic GO terms have been omitted, however they should be included in order to evaluate if there is a real enrichment of mitotic functions among regulated genes in mitotic MTA1 depleted cells.

Figure s7c. Densitometric analysis for co-ip experiment seems to over-estimate the very mild (almost absent) difference in interaction revealed by the assay. Moreover, as difference in the interaction are in authors' opinion due to differential expression of mad1 and 2 expression caused by mta2 depletion, input levels for these proteins in the different samples should be shown.

Figure s8. Gene ontology analysis for splicing-regulated genes should be carried using genes expressed by mitotic cells as background, otherwise enrichment in mitotic function could be due to higher expression in cells undergoing a mitotic switch. Has this been performed?

Figure 7 and s9. Do red and white arrows point to different defects? This should be clarified in the figure legend.

Response to Referees

Reviewers' comments:

Reviewer #1 (Remarks to the Author):

This paper reports a new role for MTA1 as a regulator of alternative splicing during mitosis. This finding has a high novelty factor as it reveals a hitherto unknown function of MTA1 and challenges the notion that mRNA processing is minimal during mitosis. Overall, these new claims are well supported by experimental data and extensive bioinformatic analysis. This paper is a prime example for the seamless integration of state of the art biological experimentation and computational analysis.

I only have some minor points of criticisms

1) the description of experiments and methods in the text and especially in the figure legends is often very sparse, which makes it difficult to follow experiments

Response: Thanks for your suggestion, we have made necessary revisions on the description of experiments and methods in the figure legends to provide more technical details.

2) What is not clear is the relationship between MTA1's ability to destabilize transcripts and to affect alternative splicing. This question is important as the authors show very profound effects of MTA1 knockout or overexpression on the abundance of mitosis related transcripts and alternative splicing. However, it is not clear whether the biological effects stem from the expression changes or from the alternative splicing. This question should be ideally addressed experimentally (although I concede that this may be difficult) or at least by discussion.

Response: Thank you for your insightful question. MTA1 indeed acts both to destabilize transcripts and to affect alternative splicing and even regulates a proportion of genes at both aspects. However, as the reviewer commented, it's really hard to make a clear conclusion about the inherent relationship between the two processes, which we suppose may be internally inseparable; We suggest that this may depend on its interacting RNPs and locations of bound motifs on genes. As to whether the biological effects of MTA1 on mitosis stem from the expression changes or from the alternative splicing, we tend to say it is most probably due to a comprehensive effect combining alternative splicing and abundance regulation of MTA1-bound transcripts, since both MTA1-bound DEG and alternatively spliced transcripts enriched on mitosis process. Moreover, we have validated that either knockdown of the genes TTK, DLG1 and SGO1 (the below figure, or Figure 5C in the revised manuscript) which were mitotically upregulated by MTA1 deletion or forced expression of the MYBL2-var2 (please refer to the reply to the next question, or Figure S8D-G in the revised manuscript) contributed to the recovery of the nocodazole-induced mitotic transition in MTA1-knockout HCT116 cells; all these support our speculation that the biological effects of MTA1 probably stems from both regulatory events.

3) In the same vein, it would strengthen the authors main conclusions if clear roles for some of the MTA1 dependent alternatively spliced transcripts could be shown. For instance, how does the alternative splicing change the function of the resulting proteins?

Response: Thanks for the constructive advice. We have showed in the manuscript that the splicing of MYBL2 pre-mRNA was switched from skipping to retention during mitosis. Here, we further validated by qPCR that there is a lower MYBL2-Var2 level during mitosis in the MTA1-ko cells (the below figure, or Figure S8D in the revised manuscript).

To explore whether the MTA1-regulated mitosis-specific MYBL2 alternative splicing contributes to MTA1 regulation of mitosis transition, we transfected the MYBL2 splicing variants (Var1 and Var2) to MTA1-knockout cells, and found using high content analysis system that recovery of MYBL2-var2 expression in MTA1-knockout HCT116 cancer cells contributes more than MYBL2-var1 in overcoming the mitotic arrest induced by nocodazole (the below figure, or Figure S8E-F in the revised manuscript).

The above findings were further supported by results from FACS analyses (the below figure, or Figure S8G in the revised manuscript). All these data suggest that MTA1 may drive the mitotic transition partially by mitotically modulating the alternative splicing pattern of MYBL2.

4) Some more analysis of the MTA1 status in tumours also would be helpful for supporting the physiological significance of the paper. For instance, is there a correlation between MTA1 expression and mutations in splice factors?

Response: As for the expressional status of MTA1 in tumors, we have performed a colon cancer tissue microarray assay, which shows that MTA1 is significantly up-regulated in colon cancer compared with its adjacent normal tissues. We did not found a significant correlation between MTA1 expression and splicing factor mutations, instead of a high correlation between MTA1 and splicing factors at the mRNA level in the majority of cancer types by TCGA data profiling (as shown in Figure S1D, S9I-J in the manuscript).

5) Fig.1E is insufficiently explained. It is unclear which antibodies were used for IP and which one for Western blotting.

Response: We are very sorry for the insufficient description. In Fig.1E, MTA1 antibody was used for IP, and all Western blotting assays were performed using respective protein antibodies. We have added the missing message to the figure legends.

6) Fig. 1G and SIM. At the resolution shown co-localisation is not apparent.

Response: Thank you for the criticism. For Fig.1G, we have represented the figure to show the clearer co-localization details;

As for the S1M, we have repeated the experiments and replaced the figure with a high-resolution picture (the below figure).

7) Fig. S7A. RANBP1 expression shows an opposite behavior compared to the other transcripts. Any explanation for that?

Response: Thanks for the good question! We have also noticed the opposite behavior of RANBP1 and tried to figure out an explanation. When looking from the cumulative curve, we can see MTA1-binding mainly cause

upregulation of bound transcripts. However, there are truly a part of bound transcripts showed an opposite downregulation effect. We attribute this phenomenon to the position where MTA1 binds or the potential partners MTA1 works with on the transcripts. The detailed mechanisms are interesting and worth further investigation. We will try to decipher this bidirectional regulating mechanism in the following research. Really thanks for the question to open a new research branch.

8) Fig. S7C. The blot hardly shows any differences, while the bar graph claims significant differences. How often was this experiment repeated? How was statistical significance calculated?

Response: The experiment was repeated for more than 3 times and statistical significance was calculated using one-way ANOVA. We have retreated and simplified the figure to show the clearer difference in MAD1-precipitated MAD2 amount (the below figure, left, or Figure S7E in the revised manuscript); moreover, we have repeatedly validated that MAD2 was less precipitated by MAD1 in MTA1-overexpressing HCT116 cells than control (the below figure, right). We also further examined the expression of MAD1 and MAD2 in MTA1-deleted and overexpressing cells and found that MAD1 and MAD2 also decreased in MTA1-overexpressing samples, which is in line with the changes at the mRNA level.

Reviewer #2 (Remarks to the Author):

The manuscript by Liu et al. shows that MTA1, a protein that was previously characterized as a chromatin modifier, is also a RNA-binding protein that modulates splicing, in particular during mitosis. The authors start from an interactomics screen where they observed and confirmed the association with many RNA-binding proteins. Then an adapted CLIP-Seq protocol was performed to identify and characterize the binding sites on mRNA molecules pointing to the modulation of alternative splicing. The authors observed a strong link to mitosis and continued to show that splicing is affected by MTA1 during mitosis. Dysregulation of MTA1 leads to aberrant chromosome segregation and instability, and promotes tumorigenesis. By looking at clinical cancer specimens, a correlation was observed.

This is an extensive manuscript which comprehensively covers the RNA-binding aspect of MTA1. The experiments appear well performed albeit somewhat difficult to follow because of limited information (see below).

Apart from the concerns that I have expressed below, I think the data are of good quality and, as the manuscript touches upon an exciting field with many outstanding questions. Publication can be supported if the comments are sufficiently addressed.

Major comments:

While it does make sense to provide the data as a single story, the manuscript becomes really bulky and sometimes difficult to follow (see figure legends comment below). I assume that some data were left out to keep the manuscript streamlined (see f.i. my comment on the fCLIP-seq protocol). This inevitably raises questions on quality of the data (while there may not be an issue with quality). Therefore, the authors should consider making separate manuscripts.

Response: Thanks so much! We had also thought about making the procedure a separate manuscript, but we were afraid that this would make the main manuscript even hard to follow. We had completed the protocol details as possible in this manuscript. We will discuss this issue with the journal and make a separate protocol manuscript according to the journal's advice.

As a general comment, it would help to provide some more information in the figure legends. The information is too limited now and figures are therefore hard to understand. Please also provide some information on the experiment that was performed to allow a good assessment of the data. Some abbreviations are also not explained. This will lead to a longer manuscript however.

Response: Thanks for the advice and sorry for the missing information. We had made completion of the information in the manuscript, including annotation of abbreviations, as we tried our best to make it simple.

While the authors provide an impressive amount of work, it would help if the initial observations relied on data that is somewhat more robust:

For the interactomics screening, the authors use two different anti-MTA1 antibodies to pull down the complexes associated to MTA1. Overlap is used as a criterion to assess quality. This is however not a valid approach to perform these experiments. Many proteins stick to the beadmatrix (or even the tube walls) and there is thus no way to make a distinction on what is specific and unspecific. More disturbingly is the fact that these sticky proteins are typically abundant proteins and many of them are binding RNA. Ribonucleoproteins belong to the most detected protein families in the CRAPome (Mellacheruvu, Nat Methods 2013) implying that there is a good chance to detect these proteins anyhow, independent of the protocol. A better approach would therefore be to include a number of control experiments in the design. Good controls are antibody controls (antibody pool from the same species but not specific for a protein target). For data analysis, either quantitative proteomics can be used (e.g. LFQ with MaxQuant and Perseus visualization), or tools such as SAINT, SFINX or others, which focus on spectral or peptide counts. The downstream validation supports (some of) the observations that were made using the ill-designed MS experiment, so it should be possible to confirm this. In addition, the mass spec data should be deposited in a repository (e.g. PRIDE).

Response: Thanks for your criticisms on the protocol design. To make up for the shortage in the design of MS experiment, we have performed downstream Co-IP and immunofluorescent co-localization analyses and further validated the specific interaction between MTA1 and RNPs, which supported the specificity of the MS experiment.

The specific interaction of MTA1 with RBP or RNA was also supported by the retracted data from the supplementary data of other studies (Hein, et al.,2015) (He, et al.,2016) (Thorenoor, et al.,2016) which was discussed in the discussion section of the manuscript.

The authors rely on an adapted protocol (fCLIP-Seq) to show the binding to RNA sites. While the data that was obtained may be true, it is risky to rely on a new protocol to make these statements. Are there other manuscripts in preparation or papers where this approach was described? It would definitely help to have some kind of positive control to show that this approach is actually performing as expected. This would help make sure that there are no unexpected artifacts that explain the observations.

Response: Thanks for your kind suggestion. We found some other studies using similar formaldehyde cross-linking method to capture the RNA associated proteins (Silverman, Li et al. 2014, Gosai, Foley et al. 2015, D, Kelley et al. 2016, Kim, Jeong et al. 2017), though we started fCLIP-seq experiments earlier than these reports. Kim et al established a protocol termed ‘‘formaldehyde crosslinking, immunoprecipitation, and sequencing (fCLIP-seq)’’ (Kim, Jeong et al. 2017), which we think can be treated a good positive control of our fCLIP-seq method.

Minor comments:

Line 67: rephrase Yes, correction has been done.

Line 105: Coupled with... rephrase this sentence Yes, correction has been done.

Line 133: ‘is always indicative’: this is a strong statement although it’s still indicative. Best rephrase Yes, correction has been done.

Line 138: Typo in ‘transcriptionally’ Yes, correction has been done.

Line 396: what is exactly meant by this statement? Is this a reference to a specific protein-RNA interaction? Best rephrase Yes, correction has been done.

Line 401 and M&M linked to this: how were these KO cells generated exactly? Were cells infected with both gRNA constructs? If lentiviral expression remains in the infected cells, off-target sites will be cleaved also.

Response: The KO cells were generated by concomitant infection of both gRNA lentivirus. As the reviewer concerned, off-target effect is really the limitation of the stable Crispr gene-knockout. However, it is very hard to eliminate the off-target effect of the Crispr-cas9 technology now, even by using only one gRNA construct and transient transfection system.

Line 423: ENCODE project reference incomplete. OK, we had made completion

Line 429: How were the high, medium and low affinity CHIP targets defined? Total divided by 3 (implying equal numbers?)

Response: The affinity intensity of MTA1 with CHIP targets was defined using a binding affinity score, which is the sum of fold enrichment (MTA1-CHIP / Input) of peaks in the gene promoter region according to the CHIP-seq data. The affinity score was equally grouped into 3 categories, which are high, medium and low.

Figure 3J does not show all results for the RT-qPCR experiments. How was the selection made on what is shown in the figure?

Response: We selected several targets for qRT-PCR validation, including 6 IR and 12 non-IR events, 16 of which (88.89%, 16/18) were consistent with the sequencing results; the changes in 12 (75%, 12/16) were significant (p-value < 0.05, t-test). We have revised the figures to show all detected events, including 12 in figure 3J (11 significant), 1 (LTK, significant) in figure S3J and 5 (0 significant) in figure S3E.

Line 530: rephrase 'we enriched'. OK, thanks. That has been done.

Line 547: were single cell clones used in these experiments? If so: can clonal variation be a problem? Also for Fig4F

Response: Thanks for the question. Actually, clones used in both experimental sets were not single cell clones. We have constructed multiple sublines with different MTA1-overexpressing levels or with EGFP-fusion tag. HCT116-OE-3 was constructed with MTA1 ORF plasmid with no EGFP tag and possessed relative higher MTA1 level than HCT116-OE-1 and 2, while HCT116-EGFP-MTA1 cell line was constructed with EGFP-MTA1 vector. All these sublines are mixed cell clones from antibiotic selection.

Fig S7C: were these data normalized? How many times performed?

Response : Thanks for the question. They were not normalized, but the actual amount of MAD1-immunoprecipitated MAD2. Experiment has been repeated for more than three times to validate that MTA1 decreased the MAD1-precipitated MAD2. We have revised and simplified the figure to highlight the difference (Figure S7E in the revised manuscript). Moreover, we have repeated this experiment for one more time here, and further validated that MAD2 was less precipitated by MAD1 in MTA1 overexpressing HCT116 cells than control. We also further examined the expression of MAD1 and MAD2 in MTA1-deleted and -overexpressing cells and found that MAD1 and MAD2 were decreased in MTA1-overexpressing cells, which was in line with their changes at the mRNA level.

Figure 6 mostly relies on data extracted from the sequencing approaches. It would be good to confirm some of the observations using RT-qPCR.

Response: Thanks for the suggestion. We have validated the mitotically regulated expression of some transcripts such as TTK, ZWILCH, DLG1 and SGO1 using RT-PCR (the below figure, or Figure S7B in the revised manuscript).

Line 872: Would it make sense to include other members of the family to assess the specificity of this observation.

Response: Thanks for the suggestion. We did have thought to include other members for confirmation. But at a second consideration and background checkup, we thought that other MTA family members (MTA2 and MTA3) may not be good control to assess the specificity of MTA1-correlated phenotype. It would not exclude the specificity of MTA1 in this observation no matter whether the correlation between other MTA family members and mitotic defectiveness was established, since we have shown that MTA1 interacts and is co-expressed with MTA2 in cancers, this certainly will confuse the interpretation of the data.

Line 894: I think 'Figure 7G' should be 'figure 7I'

Response: Yes, it's indeed should be 'figure 7I', we feel very sorry for the mistake.

References

- D, G. H., D. R. Kelley, D. Tenen, B. Bernstein and J. L. Rinn (2016). "Widespread RNA binding by chromatin-associated proteins." *Genome Biol* **17**: 28.
- Gosai, S. J., S. W. Foley, D. Wang, I. M. Silverman, N. Selamoglu, A. D. Nelson, M. A. Beilstein, F. Daldal, R. B. Deal and B. D. Gregory (2015). "Global analysis of the RNA-protein interaction and RNA secondary structure landscapes of the Arabidopsis nucleus." *Mol Cell* **57**(2): 376-388.
- Kim, B., K. Jeong and V. N. Kim (2017). "Genome-wide Mapping of DROSHA Cleavage Sites on Primary MicroRNAs and Noncanonical Substrates." *Mol Cell* **66**(2): 258-269 e255.
- Silverman, I. M., F. Li, A. Alexander, L. Goff, C. Trapnell, J. L. Rinn and B. D. Gregory (2014).

"RNase-mediated protein footprint sequencing reveals protein-binding sites throughout the human transcriptome." Genome Biol **15**(1): R3.

He C, Sidoli S, Warneford-Thomson R, et al. 2016. High-Resolution Mapping of RNA-Binding Regions in the Nuclear Proteome of Embryonic Stem Cells. Mol Cell. **64**(2): 416-430.

Hein MY, Hubner NC, Poser I, et al. 2015. A human interactome in three quantitative dimensions organized by stoichiometries and abundances. Cell. **163**(3): 712-23.

Thorenoor N, Faltejskova-Vychytilova P, Hombach S, et al. 2016. Long non-coding RNA ZFAS1 interacts with CDK1 and is involved in p53-dependent cell cycle control and apoptosis in colorectal cancer. Oncotarget. **7**(1): 622-37.

Reviewer #3 (Remarks to the Author):

Liu and colleagues report a previously unknown role in alternative splicing regulation for the chromatin-associated protein MTA1.

First, by transcriptome analysis for several primary cancers, cell lines and patient derived xenografts authors show that genes correlated with MTA1 expression are involved in pre-mRNA processing and splicing, in addition to the predicted DNA repair and transcription functions. Direct mass spectrometry analysis confirmed that RNA-binding proteins are the major components of the MTA1 interactome. Next, sequencing of the RNAs co-precipitated with MTA1 following formaldehyde its interaction with several transcripts. Consensus sequence for the 5' splice-site represent the preferential binding motif for MTA1, whose binding is indeed enriched at the exon-intron boundary. RNA-seq analysis of MTA1 depleted cells highlighted a large number of genes and splicing events whose expression is controlled by MTA1. Notably, RNAs of both transcriptionally- and splicing-regulated genes are bound by MTA1, while poor binding is found by chip-seq on their promoters, suggesting that MTA1 influences RNA processing. They also suggest that MTA1 splicing activity regulates mitotic functions in cancer cells. In line with its role in mitotic regulation of cellular transcriptome, MTA1 ablation causes severe aneuploidy and mitotic defects in cancer cell lines; moreover, MTA1 expression levels correlates with the mitotic score of primary tumors and with the tumorigenicity of cancer cells, as shown by in vivo experiments using xenografts models in nude mice.

This manuscript deals with very interesting issues, such as post-transcriptional regulation by chromatin associated proteins and coupling between transcriptome-regulation and cell-cycle control in cancer cells.

However, many of statements and conclusions are not supported by direct experiments, but rather by correlation. Some experiments lack appropriate controls and in some key aspects the study is too descriptive (i.e. lack of exhaustive validation, absence of investigation of the mechanisms by which MTA1 exerts its splicing regulation). The authors suggest a role of MTA1 in transcript destabilization, but this is not investigated in the manuscript. Is it related to splicing or independent of it? MTA1 expression correlates with that of many splicing factors and spliceosome components, some of which interact with it. Is MTA1 required for their function? Why specifically in mitosis? Many splicing factors are also associated with chromatin and released in mitosis when chromatin condense like shown for MTA1. Are these events linked? Does MTA1 modulate the recruitment of the spliceosome or other splicing factors to the target genes? In general, how would MTA1 affect splicing? they should use few model genes to investigate the mechanism of action

Response: Thank you for your valuable comments. Each valuable question can make an interesting start of a new project. Currently, little is known about mitotic pre-mRNA processing, especially AS, even less about how

pre-mRNA processing is regulated during mitosis. Here, we show that mRNA processing exists at mitosis, and MTA1, a classical CAP capable of binding RBPs and mRNAs, could specifically regulate the mRNA processing events at mitosis. As most of the findings are really novel and the study specific on mitosis phase is very hard to perform, many of the valuable concerns could not be addressed in one paper or at this stage. For instance, though MTA1 interacts and co-expresses with many splicing factors and spliceosome components, and MTA1 is involved in several RNP granules, such as SMN, YBX1 and RPS3 by colocalization analyses, we have not investigated whether MTA1 is required for their function. Moreover, why MTA1 regulates mitosis-specific functions and what is the exhaustive mechanism are all very interesting and important scientific issues but could not precisely located currently, considering the comprehensive effects of MTA1 on mitotic phenotype through so many RNPs and RNA transcripts.

More specific comments are listed below.

Major points

Figure 1. Western blot shown for co-ip experiment are of poor quality and do not show reliable interaction for many of the analyzed protein. For instance, the same amount of nono protein is found in the supernatant of mta1 and igG ip, suggesting that it has not been depleted by the co-ip. Same criticisms for co-ip treated with RNase/DNase in fig s1n, where many RBPs do not seem to associate with MTA1 in the absence of RNase. How do the authors explain this? Overall, such interactions appear extremely weak. Likewise, the immunofluorescence analysis of the association of MTA1 with actin and tubulin cytoskeleton do not appear to be conclusive, because there is no high resolution in the images shown and partial overlap is expected from the widespread distribution of cytoskeletal proteins.

Response: Thank you for your insightful questions. The deletion of MTA1-interacted proteins by co-IP from the supernatant indeed relies on its interaction intensity with MTA1, however, as the amount of its copartners at the free status is far more than that at the interacting status, even the strong interaction between HDAC2 and MTA1 could not apparently abolish HDAC2 from the supernatant. The reason that RBPs do not seem to associate with MTA1 in the absence of RNase in Figure S1 N may be due to a low-sensitivity of Western blot exposure. The interaction between MTA1 and these RBPs, as in Figure 1E, have been validated for several times in the absence of RNase. The strong interaction between MTA1 and the whole splicing complex (SMN complex) in Figure 1E is also a good evidence for MTA1 interaction with RBPs in the absence of RNase. Moreover, we have further examined again on the interaction of MTA1 with YBX1, DHX9 and HNRNPA1 as confirmation, which also showed no obvious alteration by RNase treatment (as shown below).

As the resolution of Figure S1 M is too low to show the association of MTA1 with actin and tubulin cytoskeleton, we have repeated these experiments, clearly showing the association of MTA1 with actin and tubulin cytoskeleton. The low resolution figure has been replaced in the manuscript.

Figure 2. Authors state that they combined the advantages of formaldehyde-rip and clip technique. However, the greatest advantage of using clip technique is the chance to map only direct interaction between RNA and proteins crosslinked by the UV irradiation. This advantage is not exploited by authors. By performing formaldehyde crosslinking authors allowed retrieval of both direct and indirect interactions, as in a canonical rip-seq experiment. Could the author clarify what is the advantage? Given that MTA1 interacts with multiple DNA- and RNA-binding proteins, promiscuity between direct and indirect binding is very likely when a potent crosslinker like formaldehyde is used.

Response: MTA1 is a classical chromatin-associated protein (CAP) in NuRD complex. Recent improvements in protein-RNA crosslinking technologies have defined more multifunctional proteins with dual nucleic acid specificities, many of which are typically CAPs without classical RBDs (G Hendrickson, et al.,2016, Widespread RNA binding by chromatin-associated proteins. Genome Biol. 17: 28.). Formaldehyde cross-linking was usually used in CHIP protocols for readily isolating chromatin-associated protein (CAP) complexes and recovering DNA-CAP interactions. Formaldehyde cross-linking was recently applied in a formaldehyde RNA immunoprecipitation (fRIP) technology and was proved as an optimized cross-linking method for capturing CAP-RNA interactions (G Hendrickson, et al.,2016, Widespread RNA binding by chromatin-associated proteins. Genome Biol. 17: 28.). Thus, we decide to use the formaldehyde cross-linking method to explore MTA1-RNA interactions. Nonetheless, the fRIP-Seq protocol does not include shearing RNA down to binding site resolution, despite of the high efficiency in CAP-RNA interaction pull-down. In comparison, CLIP technology is competent

to pinpoint the interaction sites at a very high resolution. So, we further combined the advantages of the fRIP and CLIP technologies and developed a formaldehyde cross-linking immuno-precipitation (fCLIP) method to identify the RNA targets bound by MTA1 complex in HCT116 cells. Similar method were reported in the later publications during we conducted the experiments. We also admit that fCLIP-seq maybe map the MTA1-RBP-RNA interaction since MTA1 interacts with lots of RBPs, which may be a widespread mechanism for CAPs to binds RNA.

Fig2b. To test if mta1 binding is enriched in a peculiar genic region, authors compared results of their clip-experiment to RNA-seq reads. It is unclear which kind of sample this latter refers to. Nevertheless, a more appropriate comparison should have been the RNA-seq of the input RNA used for the immunoprecipitation experiment. A similar consideration holds also for fig2d: what is the background considered for the analysis of MTA1 motif localization?

Response: Thanks for your comments! We used input RNA (RNA-seq) as background to analyze the enriched binding regions of MTA1. The cells used for RNA-seq and CLIP-seq came from the same batch. For RNA-seq, we extracted the polyadenylated mature RNAs from total RNAs, which can be treated as the input RNA for fCLIP-seq experiment. In this study, we concluded that MTA1 preferred to bind to nascent transcripts prior to intron splicing compared with mature transcripts. So we think this comparison is proper for our purpose. For Figure 2D, the background sequences were randomly selected from the corresponding genomic regions.

Figure 3. Only 13 splicing-events are shown and only 8 are significant. Validation should be improved and regulated exons and/or introns should be clearly indicated in the figure.

Response: We selected several targets for qRT-PCR validation, including 6 IR and 12 non-IR events, 16 of which (88.89%, 16/18) were consistent with the sequencing results; the changes in 12 (75%, 12/16) were significant (p-value < 0.05, t-test). We have revised the figures to show all detected events, including 12 in figure 3J (11 significant), 1 (LTK, significant) in figure S3J and 5 (0 significant) in figure S3E.

Figure 4. How was stable overexpression and knockdown of MTA1 achieved? Why MTA1 appears as three bands in the western blot? What is the effect of MTA1 overexpression/downregulation on the cell cycle of cells without imposing a microtubule stress with nocodazole? Another interpretation of the data is that overexpression of MTA1 slows down the cell cycle, thus reducing the percentage of cells accumulating in G2/M in the time-frame of nocodazole treatments. Also, why different aspects of the mitotic defects correlated to variations of MTA1 expression levels are investigated using different models? Do crispr cas9 knockout cells show the same “mitotic defective” phenotype?

Response: In this study, stable MTA1 overexpression was achieved by transfection of MTA1 full CDS or EGFP-MTA1-expressing plasmid. MTA1 knockdown was achieved by shRNA or Crispr-cas9 technology at different stages of the experiments. Because shRNA only deleted about 60-70% of total MTA1, to achieve a more intensive and stable deletion effect, we applied the Crispr-cas9 after this technology is reported by Feng Zhang’s lab. Similar effects were found in shRNA or Crispr-cas9-mediated MTA1 deletion, as is shown on regulation of CTTN AS model in the manuscript, as well as on mitosis transition. Here we validated that Crispr-cas9-mediated MTA1 deletion also caused a higher mitotic arrest on nocodazole treatment using high-content analysis system or flow cytometry analyses (the below figure), which is in line with the finding from MTA1 knockdown cells using shRNA.

High-content analysis on nocodazole treatment

Flow cytometry analyses on nocodazole treatment

There are multiple alternative splicing variants of MTA1, thus MTA1 appears at least three to four bands in the western blot only if the protein in the SDS gel is well separated; this phenomenon is consistent when using at least 4 different MTA1 specific antibodies and also found in other publications.

The effect of MTA1 on the cell cycle is difficult to detect for the small mitotic proportion in the absence of nocodazole. Because MTA1 overexpression was showed to promote cell proliferating rate in our and other previous studies, we don't suppose the effect of MTA1 overexpression on mitosis arrest was due to possible slowing down of the cell cycle.

Mitotic defectiveness was more frequently found in MTA1-overexpressing cells but less in MTA1-deletion cells. Similar mitosis-defective effects were found in both MTA1- and EGFP-MTA1-overexpressing HCT116 cells. Both forced mitosis transition under nocodazole-arrested condition (Figure 4 D and F).

Figure s5. Authors state that MTA1 binding induces destabilization of bound transcripts. However, no data on stabilization of transcripts is shown here. What the data show is a change in expression levels, which may be caused by several mechanisms, including indirect transcriptional mechanisms (for instance related to cell cycle progression effects) not requiring direct binding of MTA1 to the promoter region of the regulated genes. Also, the authors refer to these genes as "mitotic transcriptome". What is the overlap with the MTA1 DEG genes described in the previous figure? Are these different genes or they mainly overlap with what described before? In general, the authors conclude this paragraph stating that MTA1 controls the mitotic transcriptome by an RNA-binding mediated mechanism. However, this statement is not supported by data, as the results are only correlative.

Response: We infer that MTA1 induces destabilization of bound transcripts during mitosis because MTA1 was deprived to regulate mRNA transcription as MTA1 is free of chromatin at mitosis, moreover most of decreased mRNAs showed MTA1 binding activity. We could not provide more solid evidences because the investigation on transcript stabilization during mitosis is really difficult to perform as the mitosis phase is usually very short. Thus we indeed could not exclude the possibility of indirect transcriptional mechanisms at this stage. We prefer to confine the "mitotic transcriptome" as to the assemble of transcripts at mitosis, which includes both the transcripts of DEGs and alternatively spliced transcripts of RASGs.

Figure 5 and s7. Authors correlate the mitotic defect observed in MTA1 depleted cells with the altered oscillatory expression of some mitotic regulators, whose transcripts are bound by MTA1. However, for none of these genes an independent validation has been carried out. Validation would corroborate this observation.

Response: Thank you for your suggestion. We have validated the MTA1-exerted mitosis-specific regulation on TTK, ZWILCH, DLG1 and SGO1 by qPCR (the below figure, or Figure S7B in the revised manuscript).

Moreover, as these genes were specifically upregulated at mitosis in MTA1-deletion cells, we further knocked down these genes using shRNA method, and found that deletion of TTK, DLG1 or SGO1 partially overcome the mitotic arrest caused by MTA1 deletion, while deletion of ZWILCH display the opposite role to others (the figure below, or Figure 5C in the revised manuscript). These results suggest that MTA1 may target TTK, DLG1 and SGO1 to regulate mitosis transition.

Figure 6 and s8. The same criticism for the lack of validation is raised for the MTA1- splicing regulated mitotic genes. Moreover, splicing regulation seems independent from MTA1 binding, because exon 16 and exon 2 of the ATRX gene show MTA1 binding and presence of its consensus motif, but their splicing is not regulated. Also, as

the splicing of MYB and ATRX fluctuates in mitosis in control cells, how is MTA1 eventually regulated to be switched on and off so quickly?

Response: Thanks for the constructive advice. We have shown in the manuscript that the splicing of MYBL2 pre-mRNA was switched from skipping to retention during mitosis. Here, we further validated by qPCR that there is a specific lower MYBL2-Var2 level during mitosis in the MTA1-ko cells (the below figure, or Figure S8D in the revised manuscript).

To explore whether the MTA1-regulated mitosis-specific MYBL2 alternative splicing contributes to MTA1 regulation of mitosis transition, we transfected the MYBL2 splicing variants (Var1 and Var2) to MTA1-knockout cells, and found using high content analysis system that recovery of MYBL2-var2 expression in MTA1-knockout HCT116 cancer cells contributes more than MYBL2-var1 in overcoming the mitotic arrest induced by nocodazole for 8 h (the below figure, or Figure S8E-F in the revised manuscript).

The above finding was also supported by FACS analyses (the below figure, or Figure S8G in the revised manuscript). All these data suggest that MTA1 may drive the mitotic transition partially by mitotically modulating the alternative splicing pattern of MYBL2.

We thought that MTA1 is an essential splicing co-regulator but not a determinant in ATRX and MYBL2 pre-mRNA alternative splicing. For ATRX, exon 6 is a known splicing region that can be regulated by splicing factors and MTA1, however, no splicing events were found on exon 16 and exon 2, which could also not be

spliced by MTA1 and other splicing factors. Similarly, there are only two splicing variants in MYBL2 produced by splicing or skipping exon 3, thus, though MTA1 or other splicing factors could bind MYBL2 pre-mRNA at other regions, no splicing events could occur at these regions.

Though the mitotic phase is very short in comparison with other cell cycle phases, mitosis is a very complicated process including prophase, metaphase, anaphase and telophase. It must need very precise and quick regulation like alternative splicing that control these transition processing events. But why and how MTA1 regulates alternative splicing of MYBL2 and ATRX so quickly is very hard to answer now and beyond the scope of the manuscript. We can only propose that this may be due to a quick switch linked to association and disassociation of RBPs with mRNAs.

Figure 7. Comparison toward samples with basal level of MTA1 should be performed in order to appreciate the mitotic phenotype variations induced by modulation of MTA1 expression levels. Statistical analysis should corroborate the significance of the difference in tumorigenicity observed in the model (figure7g).

Response: Thanks for the suggestions. We have added the missing data of mitotic defectiveness, multinucleated cell and micronucleated cell proportion analyses in sample with basal level of MTA1 to the manuscript. As for figure 7g, because the sample number is so small (n = 6), we have only gave a basal description of the data.

Minor points

Figure legend of both figure 1 and s1 should be more detailed, describing for instance the cell line used for the illustrated experiment.

Response: We have added more details to the figure legends as suggested.

Figure s6. In this figure non-mitotic GO terms have been omitted, however they should be included in order to evaluate if there is a real enrichment of mitotic functions among regulated genes in mitotic MTA1 depleted cells.

Response: Thanks. For figure S6 and S8A, we have provided supplementary tables (Table S5 and S6) including all enriched GO terms.

Figure s7c. Densitometric analysis for co-ip experiment seems to over-estimate the very mild (almost absent) difference in interaction revealed by the assay. Moreover, as difference in the interaction are in authors' opinion due to differential expression of mad1 and 2 expression caused by mta2 depletion, input levels for these proteins in the different samples should be shown.

Response: We have retreated and simplified the figure to show the difference in MAD1-precipitated MAD2 (Figure S7E in the revised manuscript). Moreover, we have repeated this experiment here and further validated that MAD2 was less precipitated by MAD1 in MTA1 overexpressing HCT116 cells than control. We also further examined the expression of MAD1 and MAD2 in MAT1-deleted and overexpressing cells and found that MAD1 and MAD2 also decreased in MTA1-OE samples, which is in line with the changes at mRNA level.

Figure s8. Gene ontology analysis for splicing-regulated genes should be carried using genes expressed by mitotic cells as background, otherwise enrichment in mitotic function could be due to higher expression in cells undergoing a mitotic switch. Has this been performed?

Response: Mitotic transcriptome is a best background to enrich function of MTA1-regulated genes at mitotic cells, however, the mitotic transcriptome background is unavailable in currently commonly used GO analysis tools. Moreover, we have also performed GO analysis on MTA1-regulated genes in non-synchronized cells which also significantly enriched on mitosis-related items.

Figure 7 and s9. Do red and white arrows point to different defects? This should be clarified in the figure legend.

Response: Thanks. The red arrows indicate multinucleated cells, while white arrows indicate micronucleated cells. We have added these messages to the figure legends.

Reviewers' comments:

Reviewer #1 (Remarks to the Author):

The authors have addressed my queries satisfactorily, and I am happy to recommend acceptance of the revised paper

Reviewer #2 (Remarks to the Author):

Apart from my comment on the interactomics screen, the authors have sufficiently addressed my remarks. I really regret that they have not done some more effort on the interactomics part because this is honestly not the way to perform such an experiment. It is definitely not good policy to suggest that this is a good discovery strategy and is actually misleading. In this case, the downstream validation is not confirming MS specificity so this argument does not hold true. A parallel experiment with only beads or an aspecific antibody (isotype control) would be best. See also my previous comments why this is not a good approach. Were the MS data also uploaded to a repository? The lists provided are clearly not sufficient.

Reviewer #3 (Remarks to the Author):

The authors have partially addressed the criticisms raised to the previous version of the manuscript. Some issues are still present, but overall the work is substantially improved.

Reviewer #2 (Remarks to the Author):

Apart from my comment on the interactomics screen, the authors have sufficiently addressed my remarks. I really regret that they have not done some more effort on the interactomics part because this is honestly not the way to perform such an experiment. It is definitely not good policy to suggest that this is a good discovery strategy and is actually misleading. In this case, the downstream validation is not confirming MS specificity so this argument does not hold true. A parallel experiment with only beads or an aspecific antibody (isotype control) would be best. See also my previous comments why this is not a good approach.

Were the MS data also uploaded to a repository? The lists provided are clearly not sufficient.

Response:

Thanks so much for your valuable advice on the manuscript revision. We agree that an IgG isotype control will be stricter in the experimental design. However, redesigning and performing the experiments will take a lot of time, and especially in the fighting against the coronavirus, the lab efficiency has been largely destroyed. As the reviewer mentioned, we turn to the CRAPome database to confirm the reliability of relevant conclusions. The database is very rich in good-quality proteomic data from various experimental designs.

We retrieved all the 5 IgG-derived proteomic package (CRAPome-IgG-195~199, from HEK293 cells). First we analyzed the percentage of precipitated-RBPs by IgG in each sample to compare with those by MTA1 antibodies (as shown in table1). Also, from the literature, we luckily found one paper involving a rabbit IgG-interactome control from HCT116 with data available. The percentage of precipitated-RBPs was calculated as well (shown in table1). The IgGs in all these reports pulled down much lower percentages of RBPs among the total proteins they baited than the MTA1 antibodies in our manuscript.

To be more supportive, from the available literatures, we choose the mass spectrum proteomic data of proteins with known RNA binding function (hnRNP D, YBX1, SOX2) as positive controls, and data of proteins not much likely supposed to be RBPs (MYC, APE1, LON, PAUF) as negative controls. Their proteomic statistics were also shown in table1.

From the data in Table 1, we can see that though RBPs are indeed frequently pulled-down proteins in the mass spectrum proteomic assay, the percentages of bound RBPs are much lower than those of generally considered non-RBP proteins and known RBPs (IgG < non-RBPs < RBPs). MTA1's feature lies in the region comparable to typical RBPs, supporting the notion of MTA1 as a potential RBP.

In our manuscript, the mass spectrum proteomic assay was used as a primary guiding clue which was followed by confirmation of IgG-controlled co-IPs, immunofluorescent co-localization assays and independent RNA-binding assays. We think a false positive conclusion from the proteomic data will not be supported by the series of verification analyses.

We are very grateful for the constructive advice and discussion on the manuscript. We achieved information from the CRApome more than that and we will complete the design in the following research.

The raw mass spectrometry proteomics data have been deposited to the ProteomeXchange Consortium (<http://proteomecentral.proteomexchange.org>) via the iProX partner repository with the dataset identifier PXD018242.

<http://proteomecentral.proteomexchange.org/cgi/GetDataset?ID=PX018242>

<https://www.iprox.org/page/project.html?id=IPX0002104000>

Table 1

	samples	known gene functions	Total proteins captured by antibody	numbers of RBPs among the captured proteins	% of RBPs/total proteins captured	% of RBPs captured/RBPs annotated	reference
IgG controls	CC195-HEK293	IgG	82	15	18.29%	0.56%	Mellacheruvu et al., 2014
	CC196-HEK293	IgG	160	45	28.13%	1.67%	Mellacheruvu et al., 2014
	CC197-HEK293	IgG	107	26	24.30%	0.97%	Mellacheruvu et al., 2014
	CC198-HEK293	IgG	98	21	21.43%	0.78%	Mellacheruvu et al., 2014
	CC199-HEK293	IgG	165	42	25.45%	1.56%	Mellacheruvu et al., 2014
	HCT116-IgG	rabbit IgG		294	92	31.29%	3.42%
proteins without known RBPs and not much likely to be RBPs	MYC	transcriptional factor	33	11	33.33%	0.41%	Lu, Yang, He, & Liu, 2020
	APE1	Endodeoxyribonuclease	444	149	33.56%	5.55%	Ayyildiz et al., 2020
	LON	protease	270	24	8.89%	0.89%	Kao et al., 2015
	PAUF	no known RBP function	414	55	13.29%	2.05%	Escudero-Paniagua et al., 2019
known RBPs	hnnpD	classical RBPs	90	58	64.44%	2.16%	Kumar et al., 2015
	YBX1	classical RBPs	124	89	71.77%	3.31%	Wang et al., 2015
	SOX2	transcriptional factor with reported RBPs function	137	94	68.61%	3.50%	Fang et al., 2011
MTA1-related data	mouse anti-MTA1		228	139	60.96%	5.17%	-
	rabbit anti-MTA1		133	77	57.89%	2.87%	-
	mouse anti-MTA1 + rabbit anti-MTA1		268	150	55.97%	5.58%	-
	mouse anti-MTA1 overlapping rabbit anti-MTA1		93	66	70.97%	2.46%	-

References

- Ayyildiz, D., Antoniali, G., D'Ambrosio, C., Mangiapane, G., Dalla, E., Scaloni, A., ... Piazza, S. (2020). Architecture of The Human Ape1 Interactome Defines Novel Cancers Signatures. *Scientific Reports*, *10*(1), 1–18. <https://doi.org/10.1038/s41598-019-56981-z>
- Escudero-Paniagua, B., Bartolomé, R. A., Rodríguez, S., De los Ríos, V., Pintado, L., Jaén, M., ... Casal, J. I. (2019). PAUF/ZG16B promotes colorectal cancer progression through alterations of the mitotic functions and the Wnt/ β -catenin pathway. *Carcinogenesis*, *40*(4), 1–11. <https://doi.org/10.1093/carcin/bgz093>
- Fang, X., Yoon, J. G., Li, L., Tsai, Y. S., Zheng, S., Hood, L., ... Lin, B. (2011). Landscape of the SOX2 protein-protein interactome. *Proteomics*, *11*(5), 921–934. <https://doi.org/10.1002/pmic.201000419>
- Kao, T. Y., Chiu, Y. C., Fang, W. C., Cheng, C. W., Kuo, C. Y., Juan, H. F., ... Lee, A. Y. L. (2015). Mitochondrial Lon regulates apoptosis through the association with Hsp60-mtHsp70 complex. *Cell Death and Disease*, *6*(2), 1–11. <https://doi.org/10.1038/cddis.2015.9>
- Katzenmaier, E. M., Fuchs, V., Warnken, U., Schnölzer, M., Gebert, J., & Kopitz, J. (2019). Deciphering the galectin-12 protein interactome reveals a major impact of galectin-12 on glutamine anaplerosis in colon cancer cells. *Experimental Cell Research*, *379*(2), 129–139. <https://doi.org/10.1016/j.yexcr.2019.03.032>
- Kumar, M., Matta, A., Masui, O., Srivastava, G., Kaur, J., Thakar, A., ... Ralhan, R. (2015). Nuclear heterogeneous nuclear ribonucleoprotein D is associated with poor prognosis and interactome analysis reveals its novel binding partners in oral cancer. *Journal of Translational Medicine*, *13*(1), 1–15. <https://doi.org/10.1186/s12967-015-0637-3>
- Lu, W., Yang, C., He, H., & Liu, H. (2020). The CARM1-p300-c-Myc-Max (CPCM) transcriptional complex regulates the expression of CUL4A/4B and affects the stability of CRL4 E3 ligases in colorectal cancer. *International Journal of Biological Sciences*, *16*(6), 1071–1085. <https://doi.org/10.7150/ijbs.41230>
- Mellacheruvu, D., Wright, Z., Couzens, A. L., Lambert, J., St-denis, N., Li, T., ... Dunham, W. H. (2013). The CRAPome: a Contaminant Repository for Affinity Purification Mass Spectrometry Data. *Nat Methods*, *10*(8), 730–736. <https://doi.org/10.1038/nmeth.2557>
- Nguyen, B. T., Pyun, J. C., Lee, S. G., & Kang, M. J. (2019). Identification of new binding proteins of focal adhesion kinase using immunoprecipitation and mass spectrometry. *Scientific Reports*, *9*(1), 1–15. <https://doi.org/10.1038/s41598-019-49145-6>
- Wang, Y., Yue, D., Xiao, M., Qi, C., Chen, Y., Sun, D., ... Chen, R. (2015). C1QBP negatively regulates the activation of oncoprotein YBX1 in the renal cell carcinoma as revealed by interactomics analysis. *Journal of Proteome Research*, *14*(2), 804–813. <https://doi.org/10.1021/pr500847p>